# Post-Training Language Models for Crosslingual Consistency

**Tianyu Liu** [* 1]  **Jirui Qi** [* 2]  **Mrinmaya Sachan** [1]  **Ryan Cotterell** [1]  **Raquel Fernández** [3]  **Arianna Bisazza** [2]

## Abstract

Language models often respond inconsistently to translation-equivalent prompts across languages, undermining the reliability of multilingual systems. To quantify this, we give an information-theoretic definition of crosslingual consistency as a divergence bound between a model's response distribution and its round-trip pushforward across languages. We then introduce penalized consistency optimization (PCO), a post-training procedure that couples this divergence with a Kullback–Leibler penalty to a fixed reference language model. Because direct optimization of PCO requires expensive on-policy roll-outs, we propose a tractable surrogate, direct consistency optimization (DCO), which can be optimized off-policy. Across diverse language models and 26 languages, DCO significantly improves crosslingual consistency, outperforms existing methods, and enables targeted alignment of low-resource languages.

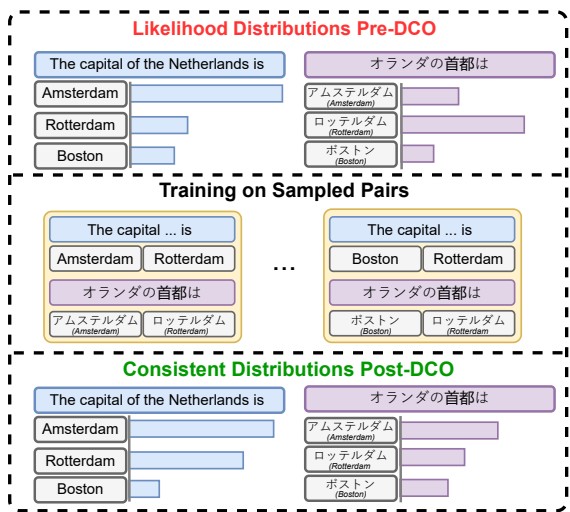

*Figure 1.* Illustration of DCO. DCO promotes crosslingual consistency by aligning the response likelihoods assigned to parallel prompts. Before alignment, these likelihoods induce inconsistent preference rankings across languages. After alignment, the ranking of candidate responses agrees in both languages.

## 1. Introduction

As multilingual capabilities become a standard feature of modern language models (LMs; Touvron et al., 2023; OpenAI et al., 2023; DeepSeek-AI et al., 2025), aligning model behavior across languages has become increasingly critical. Ideally, a language model should provide consistent responses to a prompt regardless of the language in which the model is prompted. In this paper, we call this desideratum *crosslingual consistency*. Unfortunately, crosslingual consistency is far from commonplace; indeed, prior work (Jiang et al., 2020; Qi et al., 2023; Wang et al., 2025b) has demonstrated that language models often produce conflicting responses across languages, as illustrated in Fig. 1 (top).

The lack of crosslingual consistency undermines the reliability of multilingual systems and limits their practical utility for any application that operates across languages.

In this paper, we first give a precise, information-theoretic definition of crosslingual consistency. The definition asks that a language model's response distribution in one language be close to its round-trip pushforward into that same language—the distribution obtained by translating the prompt to the other language, sampling a response there, and translating it back. Closeness is measured by an $f$-divergence. Building on this definition, we derive a new post-training objective—*penalized consistency optimization* (PCO). PCO pairs a crosslingual-consistency divergence between a language model's response distributions on translation-equivalent prompts with a Kullback–Leibler penalty that anchors the aligned model to a fixed reference. The two terms trade off crosslingual consistency against drift from the reference. We characterize the unique optimum of PCO in closed form as a weighted geometric mean of the reference language model and its round-trip pushforward. Under an invertibility assumption, we show this optimum is crosslingually consistent in the sense of our definition.

---

[*]Co-first authors [1]ETH Zürich [2]CLCG, University of Groningen [3]University of Amsterdam. Correspondence to: Tianyu Liu <tianyu.liu@inf.ethz.ch>, Jirui Qi <j.qi@rug.nl>, Ryan Cotterell <ryan.cotterell@inf.ethz.ch>.

*Proceedings of the 43rd International Conference on Machine Learning*, Seoul, South Korea. PMLR 306, 2026. Copyright 2026 by the author(s).

Direct optimization of PCO is expensive—policy gradient (Williams, 1992; Sutton et al., 1999) demands fresh on-policy roll-outs at every step and yields high-variance gradient estimates. To avoid this expense, inspired by direct preference optimization (DPO; Rafailov et al., 2023), we instead propose *direct consistency optimization* (DCO), a logit-matching supervised loss on parallel prompt pairs (Fig. 1, middle). We show that DCO's unique minimizer coincides exactly with that of PCO—so DCO recovers the same closed-form optimum without the need for on-policy roll-outs. In practice, DCO requires only offline evaluation on parallel data, making it a practical scheme for achieving crosslingual consistency.

We evaluate the effectiveness of our crosslingual consistency post-training procedure across three datasets, covering 26 languages. Experimental results demonstrate that DCO significantly improves crosslingual consistency while maintaining, and often improving, response accuracy in the post-trained languages. In addition, we show that we can control the stability of languages by using an extra strength parameter, which is useful for aligning low-resource languages with dominant ones. Together, these results indicate that DCO delivers on the consistency desideratum motivated at the outset of this section.

## 2. Preliminaries

**Language Models.** An **alphabet** is a finite, non-empty set, denoted $\Sigma$. Elements of $\Sigma$ are called **symbols**, written with lowercase Latin letters $x, y, z$. A finite sequence of symbols is called a **string**, written with bolded lowercase letters $\boldsymbol{x}, \boldsymbol{y}, \boldsymbol{z}$. We write $\Sigma^*$ for the set of all strings over $\Sigma$, including the empty string $\varepsilon$. A **language model** is a probability distribution over $\Sigma^*$, written with lowercase Greek letters such as $\pi$, $\mu$, or $\nu$. The set of all language models over $\Sigma$ is denoted $\Delta(\Sigma^*)$. A **prompted language model** $\pi \colon \Sigma^* \rightsquigarrow \Sigma^*$ is a function $\Sigma^* \to \Delta(\Sigma^*)$ assigning, to each **prompt** $\boldsymbol{x}$, a distribution $\pi(\cdot \mid \boldsymbol{x})$ over $\Sigma^*$. We use the squiggly arrow $\rightsquigarrow$ throughout to denote a prompted language model.[1] The conditional probability $\pi(\boldsymbol{y} \mid \boldsymbol{x})$ assigned by a prompted language model is the probability of generating *exactly* the **response** $\boldsymbol{y}$. Given two prompted language models $\pi, \nu \colon \Sigma^* \rightsquigarrow \Sigma^*$, the **pushforward** of $\nu$ through $\pi$ is the prompted language model $\pi \sharp \nu \colon \Sigma^* \rightsquigarrow \Sigma^*$ defined by

$$(\pi \sharp \nu)(\boldsymbol{z} \mid \boldsymbol{x}) \overset{\text{def}}{=} \sum_{\boldsymbol{y} \in \Sigma^*} \pi(\boldsymbol{z} \mid \boldsymbol{y}) \nu(\boldsymbol{y} \mid \boldsymbol{x}). \quad (1)$$

We treat $\sharp$ as a binary operator between two prompted language models, which itself returns a prompted language model. The pushforward is associative, i.e., $\pi \sharp (\nu \sharp \mu) =$

---

[1]Prompted language models can also be viewed as Markov kernels (Kallenberg, 2002) or stochastic maps (Baez & Fritz, 2014).

$(\pi \sharp \nu) \sharp \mu$, but not commutative, i.e., $\pi \sharp \nu \neq \nu \sharp \pi$ in general. We say a pair of prompted language models $(\mu, \nu)$ is **invertible** if their round-trip is the identity, $\mu \sharp \nu = \mathbf{id}$ and $\nu \sharp \mu = \mathbf{id}$.

**Annealing.** For a prompted language model $\pi$ and a temperature $T > 0$, we write $\pi_T$ for the $T$-**annealed** prompted language model obtained by raising $\pi(\cdot \mid \boldsymbol{x})$ to the $T^{\text{th}}$ power and renormalizing as follows

$$\pi_T(\boldsymbol{y} \mid \boldsymbol{x}) \overset{\text{def}}{=} \frac{\pi(\boldsymbol{y} \mid \boldsymbol{x})^T}{\sum_{\boldsymbol{y'} \in \Sigma^*} \pi(\boldsymbol{y'} \mid \boldsymbol{x})^T}. \quad (2)$$

Throughout, $\pi_T$ refers to this implicitly normalized object, so $\pi_T$ is again a prompted language model.[2]

**$f$-Divergences.** For a convex function $f \colon \mathbb{R}_{>0} \to \mathbb{R}$ with $f(1) = 0$, the $f$-**divergence** between two language models $\pi$ and $\nu$ over $\Sigma$ is

$$D_f(\pi \| \nu) \overset{\text{def}}{=} \mathbb{E}_{\boldsymbol{y} \sim \nu} \left[ f \left( \frac{\pi(\boldsymbol{y})}{\nu(\boldsymbol{y})} \right) \right]. \quad (3)$$

One canonical instance will recur throughout the paper. The **forward** Kullback–Leibler divergence is the $f$-divergence with $f(t) = t \log t$,

$$D_f(\pi \| \nu) = \mathbb{E}_{\boldsymbol{y} \sim \nu} \left[ \frac{\pi(\boldsymbol{y})}{\nu(\boldsymbol{y})} \log \frac{\pi(\boldsymbol{y})}{\nu(\boldsymbol{y})} \right] \overset{\text{def}}{=} \mathrm{KL}(\pi \| \nu), \quad (4)$$

However, note that the definition of crosslingual consistency in §3 works for any convex $f$; the PCO *objective*, however, uses reverse KL ($f(t) = -\log t$) specifically, and we verify that its optimum is consistent under any such $f$ in Prop. 2.

## 3. Crosslingual Consistency

### 3.1. The Basic Objects

We assume a single universal alphabet $\Sigma$, and identify each language with a subset of $\Sigma^*$: $L_1 \subseteq \Sigma^*$ for language one, marked in red, and $L_2 \subseteq \Sigma^*$ for language two, marked in blue. We assume access to the following distributions over $\Sigma$, all sharing the same alphabet:

- a *reference* LM $\rho \in \Delta(\Sigma^*)$; we will also use the prompted reference LM $\rho \colon \Sigma^* \rightsquigarrow \Sigma^*$ induced by $\rho$;
- response-generating prompted LMs $\pi_1$ and $\pi_2$ over $L_1$ and $L_2$, respectively;
- *prompt priors* $\mu_1 \in \Delta(L_1)$ and $\mu_2 \in \Delta(L_2)$, the distributions from which prompts in each language are drawn;
- two prompted LMs $\tau_1^2 \colon L_1 \rightsquigarrow L_2$ and $\tau_2^1 \colon L_2 \rightsquigarrow L_1$, called *translators*.

---

[2]For $T \geq 1$, the normalizer is automatically finite, since $\pi(\boldsymbol{y} \mid \boldsymbol{x})^T \leq \pi(\boldsymbol{y} \mid \boldsymbol{x})$ pointwise; for $T < 1$, we implicitly assume the sum $\sum_{\boldsymbol{y'} \in \Sigma^*} \pi(\boldsymbol{y'} \mid \boldsymbol{x})^T$ converges.

With this notation, the push-forward $\tau_1^2 \sharp \pi_1 \in \Delta(L_2)$ is the distribution over $L_2$ obtained by sampling a string in $L_1$ from $\pi_1$ and translating it via $\tau_1^2$, i.e.,

$$(\tau_1^2 \sharp \pi_1)(\boldsymbol{y}_2 \mid \boldsymbol{x}_1) = \sum_{\boldsymbol{y}_1 \in L_1} \tau_1^2(\boldsymbol{y}_2 \mid \boldsymbol{y}_1)\pi_1(\boldsymbol{y}_1 \mid \boldsymbol{x}_1). \quad (5)$$

The conditional distribution is defined by a **round-trip** $\tau_1^2 \sharp \pi_1 \sharp \tau_2^1$, corresponding to a stochastic process that first translates a prompt $\boldsymbol{x}_2$ to $\boldsymbol{x}_1$, generates the response $\boldsymbol{y}_1$, then translates it back to $\boldsymbol{y}_2$.

$$(\tau_1^2 \sharp \pi_1 \sharp \tau_2^1)(\boldsymbol{y}_2 \mid \boldsymbol{x}_2) = \qquad (6)$$
$$\sum_{\boldsymbol{x}_1, \boldsymbol{y}_1 \in L_1} \tau_1^2(\boldsymbol{y}_2 \mid \boldsymbol{y}_1)\pi_1(\boldsymbol{y}_1 \mid \boldsymbol{x}_1)\tau_2^1(\boldsymbol{x}_1 \mid \boldsymbol{x}_2).$$

Both Eq. (5) and Eq. (6) are valid probability distributions.

## 3.2. Crosslingual Consistency

We now introduce a definition for crosslingual consistency. Algebraically (Fig. 2), the property we want to formalize is that, starting from a prompt $\boldsymbol{x}_1 \in L_1$, the round-trip, translate to $\boldsymbol{x}_2$ via $\tau_1^2$, sample a response under $\pi_2(\cdot \mid \boldsymbol{x}_2)$, and back-translate via $\tau_2^1$, yields the same distribution over $L_1$ as direct sampling under $\pi_1(\cdot \mid \boldsymbol{x}_1)$. Symmetrically, the same property holds for prompts $\boldsymbol{x}_2 \in L_2$. Def. 1 below relaxes this exact commutativity to an $\varepsilon$-tolerant $D_f$-divergence bound.

**Definition 1** (Crosslingual Consistency). *A pair of language models $(\pi_1, \pi_2)$ is $(f, \varepsilon)$-**crosslingually consistent** under translators $(\tau_1^2, \tau_2^1)$ for prompt $\boldsymbol{x}_1 \in L_1$ if there exist temperatures $T_1, T_2 > 0$ such that*

$$D_f\big(\pi_1(\cdot \mid \boldsymbol{x}_1)\big\|(\tau_2^1 \sharp \pi_2 \sharp \tau_1^2)^{T_1}(\cdot \mid \boldsymbol{x}_1)\big) \leq \varepsilon, \quad (7)$$

*and, likewise, for prompt $\boldsymbol{x}_2 \in L_2$ if*

$$D_f\big(\pi_2(\cdot \mid \boldsymbol{x}_2)\big\|(\tau_1^2 \sharp \pi_1 \sharp \tau_2^1)^{T_2}(\cdot \mid \boldsymbol{x}_2)\big) \leq \varepsilon. \quad (8)$$

The existential quantification over $T_1, T_2$ absorbs a natural mismatch in sharpness. In particular, the round-trip $\tau_2^1 \sharp \pi_2 \sharp \tau_1^2$ composes three stochastic operations and is generally more diffuse than direct sampling under $\pi_1$, so we ask only that *some* per-language re-tempering brings the round-trip within $\varepsilon$ of direct sampling rather than demanding pointwise agreement. The temperatures of $\tau_1^2$ and $\tau_2^1$ are not free parameters of the definition—as we will see in Prop. 2, there is a natural choice in some cases.

In practice, $\pi_1$ and $\pi_2$ are realized by a single LM $\pi$, conditioned on two language-specific prompts that elicit responses in $L_1$ and $L_2$, respectively. Thus, crosslingual consistency reduces to an internal coherence property of $\pi$.

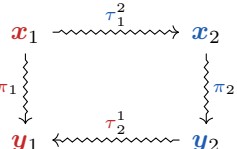

*Figure 2.* Let $\pi_1$ be a language model over $L_1$, and $\pi_2$ be a language model over $L_2$ and $\tau_1^2 \colon L_1 \rightsquigarrow L_2$ and $\tau_2^1 \colon L_2 \rightsquigarrow L_1$ be a pair of translators. The above commutativity diagram shows the case when $\tau_1^2 \sharp \pi_1 \sharp \tau_2^1 = \pi_2$ and $\tau_2^1 \sharp \pi_2 \sharp \tau_1^2 = \pi_1$, i.e., when we have $(f, 0)$-consistency for all prompts $\boldsymbol{x}_1 \in L_1$ and $\boldsymbol{x}_2 \in L_2$. For $\varepsilon > 0$, Def. 1 is a relaxation of commutativity.

## 3.3. Penalized Consistency Optimization

Let $\pi_{\boldsymbol{\theta}}$ denote the trainable language model parametrized by $\boldsymbol{\theta}$, optimized against a fixed reference $\rho$ (typically the initial checkpoint of $\pi_{\boldsymbol{\theta}}$). We also introduce two per-language strength parameters $\beta_1, \beta_2 > 0$ that weight each language's round-trip consistency reward against its unit-weight fidelity (KL) term; we revisit their interpretation in Prop. 1. We anchor the fidelity term on the same monolingual prompts as the consistency terms, so the PCO objective decomposes into two language-specific halves:

$$\mathcal{P}_1(\boldsymbol{\theta}, \boldsymbol{x}_1) \stackrel{\text{def}}{=} \underbrace{\text{KL}\big(\pi_{\boldsymbol{\theta}}(\cdot \mid \boldsymbol{x}_1)\big\|\rho(\cdot \mid \boldsymbol{x}_1)\big)}_{\text{fidelity}} - \qquad (9a)$$
$$\underbrace{\beta_1 \log(\tau_2^1 \sharp \rho \sharp \tau_1^2)(\cdot \mid \boldsymbol{x}_1)}_{\text{reward}},$$

$$\mathcal{P}_2(\boldsymbol{\theta}, \boldsymbol{x}_2) \stackrel{\text{def}}{=} \underbrace{\text{KL}\big(\pi_{\boldsymbol{\theta}}(\cdot \mid \boldsymbol{x}_2)\big\|\rho(\cdot \mid \boldsymbol{x}_2)\big)}_{\text{fidelity}} - \qquad (9b)$$
$$\underbrace{\beta_2 \log(\tau_1^2 \sharp \rho \sharp \tau_2^1)(\cdot \mid \boldsymbol{x}_2)}_{\text{reward}}.$$

and the full training objective is then given by

$$\mathcal{P}(\boldsymbol{\theta}) \stackrel{\text{def}}{=} \mathbb{E}_{\boldsymbol{x}_1 \sim \mu_1}\big[\mathcal{P}_1(\boldsymbol{\theta}, \boldsymbol{x}_1)\big] + \mathbb{E}_{\boldsymbol{x}_2 \sim \mu_2}\big[\mathcal{P}_2(\boldsymbol{\theta}, \boldsymbol{x}_2)\big], \quad (10)$$

with prompts drawn from the priors $\mu_1, \mu_2$ introduced in §3.

We now give a characterization of the minimizer under PCO.

**Proposition 1.** *Assume $\mu_1$ and $\mu_2$ are supported on $L_1$ and $L_2$, respectively, with $L_1 \cap L_2 = \emptyset$. The unique minimizer of $\mathcal{P}$ in Eq. (9a) is*

$$\pi^{\star}(\cdot \mid \boldsymbol{x}) \propto \begin{cases} (\tau_2^1 \sharp \rho \sharp \tau_1^2)^{\beta_1}(\cdot \mid \boldsymbol{x})\rho(\cdot \mid \boldsymbol{x}), & \boldsymbol{x} \in L_1, \\ (\tau_1^2 \sharp \rho \sharp \tau_2^1)^{\beta_2}(\cdot \mid \boldsymbol{x})\rho(\cdot \mid \boldsymbol{x}), & \boldsymbol{x} \in L_2, \\ \textit{any element of } \Delta(\Sigma^*), & \textit{otherwise.} \end{cases}$$
$$(11)$$

See App. C.1 for the proof. Next, we show that PCO's minimizer is crosslingually consistent.

**Proposition 2.** *Assume the exact balance condition $\beta_1\beta_2 = 1$. Further assume the translators $(\tau_1^2, \tau_2^1)$ are invertible, i.e., $\tau_1^2 \sharp \tau_2^1 = \mathrm{id}$ and $\tau_2^1 \sharp \tau_1^2 = \mathrm{id}$. Then the optimum $\pi^\star$ from Prop. 1 satisfies*

$$D_f\big(\pi^\star(\cdot \mid \boldsymbol{x}_1) \| (\tau_2^1 \sharp \pi^\star \sharp \tau_1^2)(\cdot \mid \boldsymbol{x}_1)^{\beta_1}\big) = 0 \quad and$$
$$D_f\big(\pi^\star(\cdot \mid \boldsymbol{x}_2) \| (\tau_1^2 \sharp \pi^\star \sharp \tau_2^1)(\cdot \mid \boldsymbol{x}_2)^{\beta_2}\big) = 0,$$

*i.e., $(f, 0)$-crosslingual consistency at $(\boldsymbol{x}_1, \boldsymbol{x}_2)$ in the sense of Def. 1.*

See App. C.2 for the proof.

The per-language strength parameters $\beta_1$ and $\beta_2$ in Eq. (11) interpolate between the mode of the round-trip target (as $\beta_i \to \infty$, where the round-trip target concentrates on its argmax) and $\rho$ (as $\beta_i \to 0$, where $\pi^\star \to \rho$). The balance condition $\beta_1\beta_2 = 1$ aligns the two language-specific strength parameters and, together with invertible translators, makes $\pi^\star$ exactly crosslingually consistent (Prop. 2). Setting $\beta_1 = \beta_2 = 1$ is the canonical symmetric choice; asymmetric choices satisfying $\beta_1\beta_2 = 1$, e.g., $\beta_1 = 2$, $\beta_2 = 1/2$, bias the optimum toward $\rho$ in the language with the smaller strength parameter, a knob we exploit in §5.4 to target low-resource languages without degrading the high-resource side.

### 3.4. Direct Consistency Optimization

Direct minimization of PCO is computationally expensive—the crosslingual consistency term in Eq. (9a) is an expectation under $\pi_{\boldsymbol{\theta}}$, so estimating its gradient demands on-policy roll-outs at every step. Following Rafailov et al. (2023), we sidestep this by exploiting the fact that Prop. 1 already characterizes the minimizer $\pi^\star$ in closed form, and thus opt to fit $\pi_{\boldsymbol{\theta}}$ to $\pi^\star$ as a supervised regression problem. Taking logs in Eq. (11) yields

$$\log \pi^\star(\boldsymbol{y}_1 \mid \boldsymbol{x}_1) = \beta_1 \log(\tau_2^1 \sharp \rho \sharp \tau_1^2)(\boldsymbol{y}_1 \mid \boldsymbol{x}_1) \quad (12a)$$
$$+ \log \rho(\boldsymbol{y}_1 \mid \boldsymbol{x}_1) - \log Z_1(\boldsymbol{x}_1),$$
$$\log \pi^\star(\boldsymbol{y}_2 \mid \boldsymbol{x}_2) = \beta_2 \log(\tau_1^2 \sharp \rho \sharp \tau_2^1)(\boldsymbol{y}_2 \mid \boldsymbol{x}_2) \quad (12b)$$
$$+ \log \rho(\boldsymbol{y}_2 \mid \boldsymbol{x}_2) - \log Z_2(\boldsymbol{x}_2).$$

Here, $\log Z_1(\boldsymbol{x}_1)$ and $\log Z_2(\boldsymbol{x}_2)$ are response-*independent* log-normalizers, so the two right-hand sides specify $\log \pi^\star(\cdot \mid \boldsymbol{x}_1)$ and $\log \pi^\star(\cdot \mid \boldsymbol{x}_2)$ pointwise up to those prompt-specific shifts.

To derive DCO, we parameterize the language model as

$$\pi_{\boldsymbol{\theta}}(\boldsymbol{y} \mid \boldsymbol{x}) = \frac{\exp(z_{\boldsymbol{\theta}}(\boldsymbol{y} \mid \boldsymbol{x}))}{\sum_{\boldsymbol{y}' \in \Sigma^*} \exp(z_{\boldsymbol{\theta}}(\boldsymbol{y}' \mid \boldsymbol{x}))}. \qquad (13)$$

We regress the logits $z_{\boldsymbol{\theta}}$ against the *unnormalized* log-targets $\beta_1 \log(\tau_2^1 \sharp \rho \sharp \tau_1^2) + \log \rho$ and $\beta_2 \log(\tau_1^2 \sharp \rho \sharp \tau_2^1) + \log \rho$

under a norm $\| \cdot \|$. We define the following sub-objectives

$$\mathcal{D}_1(\boldsymbol{\theta}, \boldsymbol{x}_1) \stackrel{\text{def}}{=} \sum_{\boldsymbol{y}_1 \in L_1} \Big\| \Big( z_{\boldsymbol{\theta}}(\boldsymbol{y}_1 \mid \boldsymbol{x}_1) \qquad (14a)$$
$$- \beta_1 \log(\tau_2^1 \sharp \rho \sharp \tau_1^2)(\boldsymbol{y}_1 \mid \boldsymbol{x}_1) - \log \rho(\boldsymbol{y}_1 \mid \boldsymbol{x}_1)\Big)\Big\|,$$
$$\mathcal{D}_2(\boldsymbol{\theta}, \boldsymbol{x}_2) \stackrel{\text{def}}{=} \sum_{\boldsymbol{y}_2 \in L_2} \Big\| \Big( z_{\boldsymbol{\theta}}(\boldsymbol{y}_2 \mid \boldsymbol{x}_2) \qquad (14b)$$
$$- \beta_2 \log(\tau_1^2 \sharp \rho \sharp \tau_2^1)(\boldsymbol{y}_2 \mid \boldsymbol{x}_2) - \log \rho(\boldsymbol{y}_2 \mid \boldsymbol{x}_2)\Big)\Big\|.$$

Under the invertible-translator assumption of Prop. 2 ($\tau_1^2 \sharp \tau_2^1 = \mathrm{id}$ and $\tau_2^1 \sharp \tau_1^2 = \mathrm{id}$), the translators are deterministic bijections, so each round-trip in Eq. (6) reduces to a single evaluation of $\rho$ on translated input–output pairs, and Eq. (14) can be computed efficiently in closed form. In the general case, one can fall back to Monte Carlo estimation, i.e., given a prompt $\boldsymbol{x}_1$, sample its translation $\boldsymbol{x}_2$ from $\tau_1^2$, and sample the response $\boldsymbol{y}_2$ from $\rho(\cdot \mid \boldsymbol{x}_2)$. The full DCO objective is then given by

$$\mathcal{D}(\boldsymbol{\theta}) \stackrel{\text{def}}{=} \mathop{\mathbb{E}}_{\boldsymbol{x}_1 \sim \mu_1} \big[\mathcal{D}_1(\boldsymbol{\theta}, \boldsymbol{x}_1)\big] + \mathop{\mathbb{E}}_{\boldsymbol{x}_2 \sim \mu_2} \big[\mathcal{D}_2(\boldsymbol{\theta}, \boldsymbol{x}_2)\big]. \quad (15)$$

Because we fit the logits directly, we avoid having to compute $\log Z_1(\boldsymbol{x}_1)$ and $\log Z_2(\boldsymbol{x}_2)$, both of which involve an infinite sum. Prop. 3 below makes this precise.

**Proposition 3.** *Assume $\mu_1$ and $\mu_2$ are supported on $L_1$ and $L_2$, respectively, with $L_1 \cap L_2 = \emptyset$. The minimizer $z^*$ of $\mathcal{D}$ (Eq. (15)), unique up to a prompt-dependent additive constant, is given by*

$$z^*(\boldsymbol{y} \mid \boldsymbol{x}) = \log \pi^\star(\boldsymbol{y} \mid \boldsymbol{x}), \qquad (16)$$

*with $\pi^\star$ defined in Eq. (11).*

See App. C.3 for the proof.

## 4. Experimental Setup

**Choice of norm for DCO.** In our experiments, we use the $L^1$-norm for the DCO. Preliminary experiments with $L^2$-norm performed worse—hence, our focus on the $L^1$-norm.

**Models.** We evaluate our method on 9 multilingual models from 4 LM families with sizes ranging from 3B to 14B, namely: `Qwen2.5-7B/14B` (Qwen et al., 2025), `Qwen3-8B/14B` (Yang et al., 2025), `Aya-Expanse-8B` (Üstün et al., 2024), `Llama3.1-8B`, `Llama3.2-3B` (Dubey et al., 2024), and `Gemma3-4B/12B` (Kamath et al., 2025). Details on the training configurations are provided in App. E.

**Datasets.** We consider three different multilingual question answering benchmarks: MMMLU (Hendrycks et al.,

| Method | Qwen2.5-14B | | | Gemma3-12B-pt | | | Qwen3-14B | | | Aya-Expanse-8B | | | Llama3.1-8B | | |
|---|---|---|---|---|---|---|---|---|---|---|---|---|---|---|---|
| | $CLC_{all}$ | $A_{EN}$ | $A_{\neg EN}$ | $CLC_{all}$ | $A_{EN}$ | $A_{\neg EN}$ | $CLC_{all}$ | $A_{EN}$ | $A_{\neg EN}$ | $CLC_{all}$ | $A_{EN}$ | $A_{\neg EN}$ | $CLC_{all}$ | $A_{EN}$ | $A_{\neg EN}$ |
| Base | 68.6 | 72.5 | 58.1 | 73.6 | 70.1 | 62.3 | 76.1 | 76.6 | 67.3 | 72.2 | 59.8 | 52.9 | 60.9 | 57.3 | 45.8 |
| + SFT* | +0.6 | +1.5 | +6.7 | +0.6 | +0.7 | +1.6 | −0.2 | +0.1 | +0.5 | +3.5 | +0.7 | +0.5 | +4.3 | +6.7 | +5.9 |
| + DPO* | +12.3 | +7.8 | +13.9 | +6.5 | +1.8 | +3.4 | +3.0 | +2.7 | +4.2 | +1.3 | +2.5 | +2.5 | +10.1 | +8.0 | +8.8 |
| + DCO* | +13.1 | +7.6 | +13.5 | +10.2 | +1.2 | +2.9 | +4.4 | +2.8 | +4.3 | +3.1 | +2.7 | +2.6 | +13.8 | +7.3 | +8.9 |
| + CALM | +4.2 | +0.0 | +4.1 | +3.0 | −0.4 | −0.0 | +0.3 | −2.1 | −1.1 | +1.4 | −2.2 | −2.1 | +3.0 | −2.2 | −5.0 |
| + DCO | +10.6 | +4.0 | +9.6 | +6.5 | +0.9 | +2.5 | +2.7 | +0.4 | +1.3 | +5.3 | +0.5 | +0.5 | +9.4 | +7.5 | +7.6 |

*Table 1.* Comparison with prior methods on MMMLU under the joint-training setup. Methods marked with * are trained on human-annotated prompt–response pairs. See Table 8 for the full results.

2021), XCSQA (Lin et al., 2021), and BMLAMA (Qi et al., 2023). All three contain parallel prompts and candidate responses over all tested languages, translated from their English origin. MMMLU is a multilingual extension of the MMLU dataset on *general knowledge*, translated into 14 languages by human annotators. LMs are given a prompt and four candidate responses, and have to generate one option from $\{A, B, C, D\}$. In XCSQA, prompts are also multi-choice (5 options, 16 languages), but focus on *commonsense reasoning*. By contrast, BM-LAMA (Qi et al., 2023) evaluates LMs' *factual associations* in 17 languages via parallel sentence prefixes paired with a varying number of candidate responses—for example, the prefix "The capital of Italy is" with responses "{Rome, Paris, ...}". More detailed statistics and examples are provided in App. F.

**Evaluation Metrics.** We report two metrics. (i) *Crosslingual consistency* is approximated by RankC (Qi et al., 2023), which compares the likelihood of candidate responses across languages.[3] (ii) *Response accuracy* follows the LM-Evaluation-Harness protocol (Gao et al., 2024): the candidate completion with the highest model likelihood is selected and compared against the reference completion.[4] With $CLC_{all}$, we denote the average crosslingual consistency (measured by RankC) between all language pairs as a general metric of crosslingual consistency of an LM. $A_{EN}$ and $A_{\neg EN}$ are the average accuracy on English and non-English instances, respectively.

[3] We use RankC as the main metric instead of directly computing the divergence in Def. 1 because the latter requires summation over the full generation space, which is intractable in practice. RankC operates only on the parallel candidate responses provided by the benchmark, is ranking-based (thus insensitive to tokenization differences and absolute likelihood scale across languages), and is the standard metric in prior work (Qi et al., 2023), enabling direct comparison. See App. G for the precise definition of RankC.

[4] https://github.com/EleutherAI/lm-evaluation-harness

**Baselines.** We compare DCO with three previously proposed methods: supervised fine-tuning (SFT), DPO, and *crosslingual self-aligning ability of language models* (CALM; Wang et al., 2025b). In the case of SFT, we fine-tune the LMs on the prompts paired with their reference completions. In the case of DPO, we construct preference pairs using the reference completion as the preferred responses, and a randomly sampled wrong response as the dispreferred response. We also evaluate two-stage approaches where the model is first trained with DPO on reference completions and then refined with DCO using the *same* prompts used for DPO. CALM is an unsupervised method. It constructs preference pairs without the need for supervision: for each prompt, it queries the model in multiple languages, takes the majority-voted response across languages as the preferred completion, and pairs it against the model's original response in each language if they differ. It then runs DPO on these pairs, with the reference completion as the preferred responses.

## 5. Results

### 5.1. Multilingual Experiments

Following the setup of Wang et al. (2025b), we first experiment on a multilingual post-training setup where each LM is aligned across $N$ languages jointly in a single post-training process. In this setup, we experiment on $N = 12$ languages on the general knowledge dataset MMMLU.[5]

Table 1 shows the experimental results. SFT produces modest improvements in $CLC_{all}$ and response accuracy on Llama3.1-8B. However, on larger models, e.g., Qwen3-14B, the improvement in $CLC_{all}$ is negligible. We interpret this finding as indicating that SFT alone is insufficient to improve crosslingual consistency. In contrast, DPO

[5] We exclude Bengali as its input length exceeds the capacity of four A100 GPUs, and Swahili and Yoruba because most of the experimented LMs perform at random chance in these languages. See §5.4 for targeted experiments on the low-resource languages, which demonstrate promising results with adjusted $\beta_i$ values.

| Model | MMMLU | | | XCSQA | | | BMLAMA | | |
|---|---|---|---|---|---|---|---|---|---|
| | $CLC_{all}$ | $A_{EN}$ | $A_{\neg EN}$ | $CLC_{all}$ | $A_{EN}$ | $A_{\neg EN}$ | $CLC_{all}$ | $A_{EN}$ | $A_{\neg EN}$ |
| Qwen2.5-14B | 68.6 | 72.5 | 58.1 | 64.6 | 87.0 | 56.9 | 41.9 | 62.7 | 38.6 |
| + DCO | +12.6 | +1.6 | +8.5 | +6.8 | −2.5 | +4.7 | +15.4 | +6.3 | +14.2 |
| Gemma3-12B-pt | 73.6 | 70.1 | 62.3 | 58.3 | 66.0 | 47.2 | 42.2 | 68.3 | 38.3 |
| + DCO | +7.2 | −0.9 | +1.4 | +4.6 | +0.1 | +3.6 | +16.7 | +1.6 | +17.0 |
| Qwen3-14B | 76.1 | 76.6 | 69.0 | 61.9 | 77.6 | 54.0 | 38.9 | 58.4 | 36.4 |
| + DCO | +4.8 | +0.1 | +1.7 | +7.1 | +1.1 | +3.8 | +16.2 | +8.1 | +14.5 |
| Aya-Expanse-8B | 72.2 | 59.8 | 52.9 | 62.6 | 78.0 | 54.4 | 41.9 | 67.0 | 37.8 |
| + DCO | +5.3 | +0.5 | +0.5 | +6.4 | +0.6 | +3.7 | +12.3 | +1.4 | +12.2 |
| Llama3.1-8B | 60.9 | 57.3 | 45.8 | 60.2 | 67.5 | 47.7 | 40.9 | 61.2 | 35.8 |
| + DCO | +12.1 | +0.7 | +3.0 | +9.1 | +0.2 | +1.3 | +15.7 | +7.2 | +17.6 |

*Table 2.* Crosslingual consistency with English, together with average accuracy on English and non-English instances, on MMMLU, XCSQA, and BMLAMA. See App. K for the full results across nine LMs.

yields greater gains in $CLC_{all}$, $A_{EN}$, and $A_{\neg EN}$ across all LMs considered.

CALM requires more than two languages, limiting its applicability in bilingual settings. Moreover, including low-resource languages can make majority voting unreliable, because, in Swahili and Yoruba, most experimented LMs perform at the level of random chance. Our empirical findings bear out this concern: CALM yields only modest $CLC_{all}$ gains, while both English and non-English accuracy often remain flat or decline. This pattern is consistent with noisy majority voting: when low-resource languages are included, their near-random predictions can corrupt the pseudo-labels used for CALM training. By contrast, DCO yields higher $CLC_{all}$ on all tested models while preserving accuracy or even improving it in many cases. Notably, on some models (e.g., Aya-Expanse-8B) DCO even surpasses DPO on $CLC_{all}$, and on the rest it nearly matches DPO without the use of reference completions. We provide detailed CLC results for all language pairs in App. J, showing that DCO improves CLC for typologically similar languages as well as for typologically dissimilar languages.

Additionally, we evaluate a two-step post-training recipe that combines reference-completion preference learning with consistency optimization (DPO+DCO). This yields the best results: applying DCO as a post-step to a DPO-trained model consistently achieves the highest $CLC_{all}$ across all language models. Accuracy remains comparable to DPO, with minor trade-offs in English for some models, and slight gains in non-English languages for others. Taken together, our results highlight DCO as the most versatile and practical option. It offers a robust path to crosslingual consistency while preserving (and often improving) task accuracy, and it can also serve as an effective post-step that further benefits models already trained with DPO.

### 5.2. Bilingual Experiments

The experiments in §5.1 target *joint* consistency improvements across many languages, a setting aligned with large multilingual foundation models. In other practical scenarios, however, developers may be interested in aligning knowledge between English and a specific target language. To assess this use case, we instantiate a *bilingual* version of DCO that aligns English with one non-English language. Our choice of English is empirically grounded: querying the LM in English yields the highest accuracy in our preliminary study (App. K). Nevertheless, we still provide alignment results between two non-English languages in App. L. We also evaluate on XCSQA (commonsense reasoning) and BMLAMA (factual association). Table 2 reports CLC and average response accuracy on English and non-English. For space reasons, we present the largest model in each family; see App. K for full results on nine LMs.

Overall, DCO substantially improves both $CLC_{all}$ and accuracy across datasets. On MMMLU, $CLC_{all}$ increases by +4.79 to +12.60 across all models, with concurrent gains in the accuracy of non-English $A_{\neg EN}$(+0.46 to +8.49) and remains largely stable in English accuracy. On XCSQA, $CLC_{all}$ improves by +4.61 to +9.10, with smaller changes in English accuracy (–2.53 to +1.07, with the single notable dip on Qwen2.5-14B), while non-English accuracy increases consistently (+1.27 to +4.67). The largest gains appear on BMLAMA, where $CLC_{all}$ improves by +12.29 to +16.65 and both English accuracy (+1.43 to +8.07) and non-English accuracy (+12.16 to +17.62) rise markedly. We hypothesize that BMLAMA benefits more from DCO because outputs are concrete factual entities rather than abstract option labels, making distributional alignment across languages more direct.

| Method | Anatomy | | | Medical Genetics | | | High School Maths | | | College Maths | | |
|---|---|---|---|---|---|---|---|---|---|---|---|---|
| | $\text{CLC}_{\text{all}}$ | $A_{\text{EN}}$ | $A_{\neg\text{EN}}$ | $\text{CLC}_{\text{all}}$ | $A_{\text{EN}}$ | $A_{\neg\text{EN}}$ | $\text{CLC}_{\text{all}}$ | $A_{\text{EN}}$ | $A_{\neg\text{EN}}$ | $\text{CLC}_{\text{all}}$ | $A_{\text{EN}}$ | $A_{\neg\text{EN}}$ |
| Base | 59.49 | 68.89 | 46.30 | 70.38 | 82.00 | 63.79 | 67.83 | 53.70 | 43.23 | 64.25 | 57.00 | 37.50 |
| + DCO | +10.94 | +1.80 | +3.76 | +10.33 | +5.43 | +7.00 | +11.27 | +2.38 | +6.80 | +11.36 | -0.36 | +12.36 |

*Table 3.* Cross-domain performance on `Qwen2.5-14B`. The model is trained with DCO on *high school microeconomics* (390 prompts) and tested on distinct domains on MMMLU. Detailed results by language, and for more test domains, are shown in App. K.

### 5.3. Out-of-Domain Generalizability

To further assess whether the benefits of DCO extend beyond the specific domain the model was post-trained on, we conduct a controlled experiment using `Qwen2.5-14B`: DCO is performed on a single subject within MMMLU (i.e., *high school microeconomics*) and evaluated on various other subjects from the same dataset. For easier interpretation of the results and for managing computational costs, we keep the bilingual DCO setup in the rest of our experiments.

Table 3 shows strong out-of-domain transfer from a single training subject. DCO increases $\text{CLC}_{\text{all}}$ by about 11% on average across all target domains, indicating that CLC is enhanced beyond the specific post-training domain. Non-English accuracy also improves significantly, with the largest gain on *college mathematics* (+12.36), reflecting effective knowledge transfer from English to lower-resourced languages, even without reference completion supervision. English accuracy is largely preserved, with only a negligible decrease of 0.36 on *college mathematics*, showing that, in this case, DCO does not overfit to the training subject or degrade the model's primary language competence. Taken together, our results support the application of DCO in the case where training data are scarce or domain-specific, and the target domain may differ from that of the training data.

### 5.4. Strength Parameter Study

**Setup.** We study how DCO's parameters $\beta_1, \beta_2$ affect the results. We focus on the low-resource transfer case, and select $L_1$ to be English and $L_2$ to be either Swahili (SW) or Yoruba (YO), which have the lowest baseline accuracy on MMMLU. We consider three settings of $\beta_1$ and $\beta_2$: **Default** ($\beta_1$=1, $\beta_2$=1), **SW/YO-Anchored** ($\beta_1$=10, $\beta_2$=0.1), and **EN-Anchored** ($\beta_1$=0.1, $\beta_2$=10). All other experimental settings remain the same as in §5.2. Note that all settings of $\beta_1$ and $\beta_2$ keep the invariant that $\beta_1\beta_2 = 1$.

**Results.** We only present results of DCO for aligning EN and SW in Fig. 3 here and leave the results between EN and YO to App. H. However, across both of SW and YO, we observe similar trends. With the default strength parameters, the EN accuracy declines by 4.80 points while SW gains +2.29. As expected, the SW-anchored strengths ($\beta_1 = 10, \beta_2 = 0.1$) severely harm accuracy for EN

prompts ($-11.74$) while minimally improving accuracy on SW ($+0.18$). Finally, the EN-anchored strengths ($\beta_1 = 0.1, \beta_2 = 10$) yield improvement in both consistency and accuracy: the accuracy of SW increases by $+4.24$ while the accuracy in EN remains stable; in this case, it even slightly increases. By contrast, CLC improves in all strength parameter settings. The baseline $\text{CLC}_{\text{all}}$ of 49.37 rises to 59.21, 59.50, 56.59 under $(\beta_1{:}\beta_2) = (1{:}1), (10{:}0.1), (0.1{:}10)$ respectively, showing that DCO reliably optimizes crosslingual consistency measured by RankC across different direction weights. Notably, the larger $\text{CLC}_{\text{all}}$ boost from the default strengths comes with a substantial EN accuracy drop, whereas the EN-anchored setup achieves a more favorable balance: that is, slightly smaller yet still significant gains in $\text{CLC}_{\text{all}}$ but stable or improved EN accuracy.

**Analysis.** To better understand the effects of $(\beta_1, \beta_2)$, we measure the proportion of prompts for which the model's response changes after post-training with DCO. As shown in Fig. 3 (right), the low-resource language exhibits a larger increase in the proportion of prompts that change than EN, and the strength parameters control which side is allowed to move. The EN-anchored setting changes only 18.73% of EN responses but 54.20% of SW. Under the default strengths, the number of responses in EN that change increases, whereas the number of responses in SW that change decreases. In the SW-anchored setting, the share of updates shifts toward EN (33.97% EN vs. 44.84% SW). A similar trend holds for English–Yoruba (see App. H). These align with the accuracy-consistency results: EN-anchored settings keep the high-resource channel stable while DCO primarily revises the low-resource outputs, thereby yielding improved CLC and higher non-EN accuracy. By contrast, low-resource anchored setups tend to induce unnecessary changes to EN outputs, while still confirming that the strength parameters can effectively control the direction and magnitude of updates. Importantly, this should not be misinterpreted as meaning that assigning a larger $\beta_i$ to EN is universally beneficial. We provide practical guidance on choosing weights for real-world applications in App. I.

### 5.5. Directly Optimizing PCO

Beyond the multiple-choice evaluations of the previous sections, we test whether the same consistency-driven reward

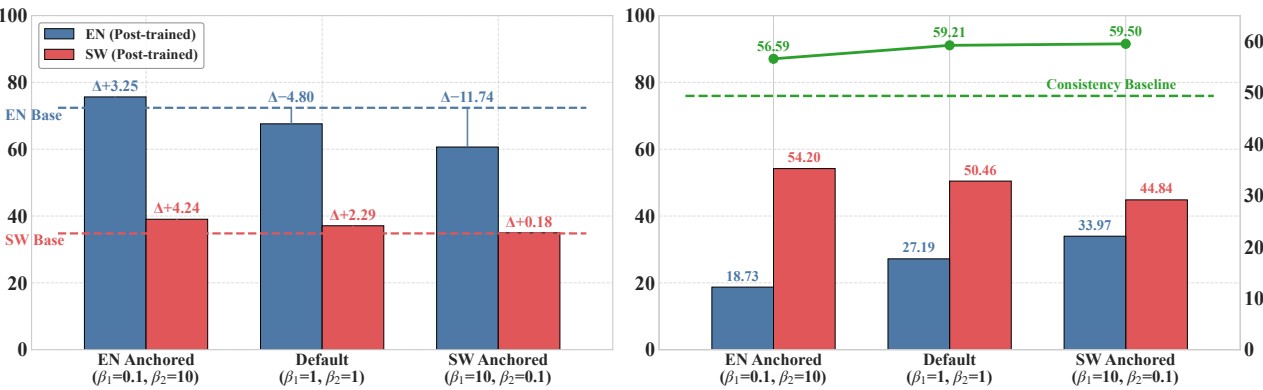

*Figure 3.* Left: Response accuracy after performing DCO on English–Swahili. Right: Proportion of prompts for which the LM's response changes after DCO, with CLC values marked in green.

can be used in an on-policy RL setting for open-ended generation. Due to compute constraints, this exploration is intentionally limited in scope: we focus on English–Chinese (EN-ZH) and two open-ended benchmarks, MMMLU and GSM8K. For MMMLU (multiple-choice), we allow the model to produce intermediate reasoning before outputting the final option; for GSM8K (multi-step math), the model generates a solution trace and then the final numeric response. To quantify CLC in this open-ended setting, we measure the overlap of correctly solved prompts across languages using the Jaccard similarity between the sets of correctly answered items in EN and ZH.

**Setup.** We optimize the PCO objective in Eq. (9a) using the algorithm of Guo et al. (2024). Each training batch contains 32 prompts sampled uniformly from the dataset. We use AdamW with learning rate 5e-6 for Qwen and 2e-6 for gemma. We emphasize that these settings are chosen for a lightweight feasibility check rather than an exhaustive hyperparameter study, and we leave a more comprehensive investigation of on-policy alignment for future work.

**GSM8K.** Without any human-annotated parallel supervision, optimizing PCO improves both per-language accuracy and EN–ZH consistency on two base models. These results suggest that DCO-style alignment generalizes to open-ended reasoning by combining translation-based pseudo pairs with a consistency-aware reward.

**MMMLU.** We observe a similar pattern on MMMLU under RL training (Table 5): optimizing our reward produces consistent gains in both accuracy and CLC. These preliminary results indicate that the approach can remain effective even when crosslingual pairs are not explicitly predefined, but we emphasize that this section is meant as an initial exploration rather than a comprehensive evaluation.

### 5.6. Discussion

**Where do the accuracy gains come from?** In general, when a language model performs poorly on a task, it tends to have a *high-entropy distribution* over the candidate responses. In contrast, a low-entropy one that is skewed toward an incorrect response is less common. A low-entropy expert only contributes minimally to the final distribution, allowing the ensemble to rely more on high-confidence predictions from other experts. As a case study, we verify this assumption using Qwen2.5-14B on the MMMLU dataset, where the average entropy of the response distribution on the prompts that are correctly answered is $0.41 \pm 0.41$, while for incorrectly answered prompts, it is significantly higher at $0.98 \pm 0.33$. The accuracy of the theoretical optimal policy $\pi^\star$ on MMMLU using Qwen2.5-14B is 77.0, surpassing the accuracy of the base policy in individual languages.

**Beyond crosslingual consistency.** While this work focuses on *crosslingual* knowledge consistency, the training objective of DCO is not limited to this task and can naturally be extended to other forms of consistency where suitable maps $\tau_2^1$ and $\tau_1^2$ can be defined. For instance, by aligning the output distributions over candidate responses of paraphrased prompts, the model could be encouraged to respond consistently regardless of surface variations.

## 6. Related Work

Several lines of work have studied crosslingual knowledge consistency in multilingual LMs, both by measuring it and by trying to improve it. On the evaluation side, alongside the RankC metric (Qi et al., 2023), Xing et al. (2024) and Ai et al. (2025) assess consistency through the agreement of top-1 responses to the same prompt posed in different languages. Jiang et al. (2020) compute the average overlap of correct predictions across languages. These studies

| Model | Acc EN | Acc ZH | Consistency |
|---|---|---|---|
| `Qwen2.5-7B-Instruct` | 89.2 | 83.6 | 84.7 |
| + on-policy RL | **90.0** | **86.8** | **86.9** |
| `gemma-3-12b-it` | 90.3 | 87.1 | 87.7 |
| + on-policy RL | **92.3** | **88.1** | **89.2** |

*Table 4.* On-policy results on GSM8K.

| Model | Acc EN | Acc ZH | Consistency |
|---|---|---|---|
| `Qwen2.5-7B-Instruct` | 66.2 | 58.8 | 68.9 |
| + on-policy RL | **70.1** | **59.7** | **72.3** |
| `gemma-3-12b-it` | 71.9 | 64.8 | 70.9 |
| + on-policy RL | **72.4** | **66.7** | **71.8** |

*Table 5.* On-policy results on MMMLU.

reveal substantial inconsistencies across a wide range of LMs. On the modeling side, a recent line of work attempts to improve consistency through vector interventions on the hidden representations of LMs (Lu et al., 2025; Wang et al., 2025a; Liu & Niehues, 2025); while promising, such methods have been validated only on small datasets and a handful of models, limiting their scalability.

## 7. Conclusion

We formalize crosslingual consistency as a divergence between a language model's response distribution and its round-trip pushforward across languages. Additionally, we introduce penalized consistency optimization (PCO), which combines this divergence with a Kullback–Leibler penalty to a fixed reference policy. Because directly optimizing PCO requires expensive on-policy roll-outs, we propose DCO, an off-policy surrogate whose unique minimizer coincides with that of PCO.

Across extensive experiments, we find DCO consistently improves CLC over a wide range of models and datasets. In our comparisons, DCO delivers robust gains over existing methods, and, when reference completions are available, combining DCO with DPO yields the strongest overall alignment in the joint $N$-language training setting. In bilingual settings, DCO likewise improves CLC, raising accuracy in non-English languages while preserving accuracy in English. DCO also generalizes across domains, with gains persisting on subjects unseen during training. Our analysis of the direction-controlling weights further shows how practitioners can steer alignment toward specific languages to meet deployment needs. The structured reward underlying DCO has potential applications beyond crosslingual knowledge consistency—for example, improving self-consistency across paraphrases (Wu et al., 2025) or consistency across modalities. Because DCO is computationally efficient, it offers a practical path to multilingual LMs that are not only accurate but also reliable and equitable across languages.

## Impact Statement

This paper advances the field of machine learning by improving the crosslingual consistency of language models. While such consistency is desirable for fairness and reliability, we caution that enforcing it too strictly risks erasing legitimate language-specific and culture-specific variation, where the "correct" answer may genuinely differ across linguistic contexts. We encourage future work to distinguish undesirable inconsistency from meaningful cultural divergence, and to treat the latter with care.

## Acknowledgments

This work was supported as part of the "Swiss AI initiative" by a grant from the Swiss National Supercomputing Center (CSCS) under project ID 87 on Alps Cluster. This work also used the Dutch national e-infrastructure with the support of the SURF Cooperative using grant no. EINF-17648.

The authors have received funding from the Dutch Research Council (NWO): JQ is supported by NWA-ORC project LESSEN (grant nr. NWA.1389.20.183). AB is supported by the above as well as NWO Talent Programme (VI.Vidi.221C.009).

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

# Appendix

## A. Language Model Usage

LM tools were used to improve writing clarity. They did not contribute to the conceptual development, experimental design, or analysis. The authors reviewed and edited all assisted text, and the final manuscript is *entirely* author-written.

## B. Discussion on the Definition of Crosslingual Consistency

One might attempt to use a stricter definition of consistency, such as requiring exact probability matches $\pi_1(\boldsymbol{y}_1 \mid \boldsymbol{x}_1) = \pi_2(\tau_1^2(\boldsymbol{y}_1) \mid \tau_1^2(\boldsymbol{x}_1))$ for every $\boldsymbol{x}_1 \in L_1$ and $\boldsymbol{y}_1 \in L_1$. Previous work has shown that even semantically identical sentences in different languages can have likelihoods that differ significantly due to lexical, semantic, and tokenization differences (Lesci et al., 2025). For instance, in `Gemma3-12B-it`, using a temperature of 1, $\pi(\text{`` Paris.''} \mid \text{``The capital of France is''}) = 0.8991$, while $\pi(\text{`` Paris.''} \mid \text{``Die Hauptstadt Frankreichs ist''}) = 0.1283$. Thus, enforcing exact probability matches would ignore these inherent differences and lead to suboptimal alignment. Instead, we adopt the softer divergence-based relaxation of Def. 1, which tolerates an $\varepsilon$-bounded $f$-divergence between $\pi_1$ and the back-translated $\tau_2^1 \sharp \pi_2 \sharp \tau_1^2$ (and vice versa) rather than pointwise equality.

## C. Proofs

### C.1. Proof of Prop. 1

**Proposition 1.** *Assume $\mu_1$ and $\mu_2$ are supported on $L_1$ and $L_2$, respectively, with $L_1 \cap L_2 = \emptyset$. The unique minimizer of $\mathcal{P}$ in Eq.* (9a) *is*

$$\pi^\star(\cdot \mid \boldsymbol{x}) \propto \begin{cases} (\tau_2^1 \sharp \rho \sharp \tau_1^2)^{\beta_1}(\cdot \mid \boldsymbol{x})\rho(\cdot \mid \boldsymbol{x}), & \boldsymbol{x} \in L_1, \\ (\tau_1^2 \sharp \rho \sharp \tau_2^1)^{\beta_2}(\cdot \mid \boldsymbol{x})\rho(\cdot \mid \boldsymbol{x}), & \boldsymbol{x} \in L_2, \\ any\ element\ of\ \Delta(\Sigma^*), & otherwise. \end{cases} \tag{11}$$

*Proof.* By Eq. (10), $\mathcal{P}$ is a $\mu_1$- and $\mu_2$-weighted expectation of per-prompt losses, so it suffices to minimize the per-prompt loss separately at each $\boldsymbol{x} \in \mathrm{supp}(\mu_1) \cup \mathrm{supp}(\mu_2)$.

**Case $\boldsymbol{x} \in L_1$.** By the disjoint-support assumption, $\mu_2(\boldsymbol{x}) = 0$, so only $\mathcal{P}_1(\boldsymbol{\theta}, \boldsymbol{x})$ contributes at $\boldsymbol{x}$. Expanding the Kullback–Leibler divergence and the cross-entropy in Eq. (9a),

$$\mathcal{P}_1(\boldsymbol{\theta}, \boldsymbol{x}) = \mathbb{E}_{\boldsymbol{y} \sim \pi_{\boldsymbol{\theta}}} \left[ \log \pi_{\boldsymbol{\theta}}(\boldsymbol{y} \mid \boldsymbol{x}) - \beta_1 \log(\tau_2^1 \sharp \rho \sharp \tau_1^2)(\boldsymbol{y} \mid \boldsymbol{x}) - \log \rho(\boldsymbol{y} \mid \boldsymbol{x}) \right]$$

$$= \mathrm{KL}\left( \pi_{\boldsymbol{\theta}}(\cdot \mid \boldsymbol{x}) \,\Big\|\, \frac{(\tau_2^1 \sharp \rho \sharp \tau_1^2)(\cdot \mid \boldsymbol{x})^{\beta_1} \rho(\cdot \mid \boldsymbol{x})}{Z_1(\boldsymbol{x})} \right) - \log Z_1(\boldsymbol{x}),$$

where $Z_1(\boldsymbol{x}) \stackrel{\mathrm{def}}{=} \sum_{\boldsymbol{y} \in L_1} (\tau_2^1 \sharp \rho \sharp \tau_1^2)(\boldsymbol{y} \mid \boldsymbol{x})^{\beta_1} \rho(\boldsymbol{y} \mid \boldsymbol{x})$ is the $\pi_{\boldsymbol{\theta}}$-independent normalizer. The Kullback–Leibler divergence is uniquely minimized in $\pi_{\boldsymbol{\theta}}(\cdot \mid \boldsymbol{x})$ at the target, yielding

$$\pi^\star(\cdot \mid \boldsymbol{x}) \propto (\tau_2^1 \sharp \rho \sharp \tau_1^2)(\cdot \mid \boldsymbol{x})^{\beta_1} \rho(\cdot \mid \boldsymbol{x}), \tag{17}$$

the $\boldsymbol{x} \in L_1$ branch of Eq. (11).

**Case $\boldsymbol{x} \in L_2$.** Symmetric: $\mu_1(\boldsymbol{x}) = 0$, and the same calculation applied to $\mathcal{P}_2$ gives

$$\pi^\star(\cdot \mid \boldsymbol{x}) \propto (\tau_1^2 \sharp \rho \sharp \tau_2^1)(\cdot \mid \boldsymbol{x})^{\beta_2} \rho(\cdot \mid \boldsymbol{x}), \tag{18}$$

the $\boldsymbol{x} \in L_2$ branch.

**Case $\boldsymbol{x} \notin L_1 \cup L_2$.** Both $\mu_1(\boldsymbol{x}) = \mu_2(\boldsymbol{x}) = 0$, so the per-prompt contribution to $\mathcal{P}$ vanishes for every choice of $\pi_{\boldsymbol{\theta}}(\cdot \mid \boldsymbol{x})$; $\pi^\star(\cdot \mid \boldsymbol{x})$ is undetermined, the third branch.

Uniqueness on $\mathrm{supp}(\mu_1) \cup \mathrm{supp}(\mu_2)$ follows from strict convexity of $\mathrm{KL}(\cdot \| p)$ in its first argument when the target $p$ has positive support on $L_i$, which $\rho$ ensures. ∎

## C.2. Proof of Prop. 2

**Proposition 2.** *Assume the exact balance condition $\beta_1 \beta_2 = 1$. Further assume the translators $(\tau_1^2, \tau_2^1)$ are invertible, i.e., $\tau_1^2 \sharp \tau_2^1 = \mathbf{id}$ and $\tau_2^1 \sharp \tau_1^2 = \mathbf{id}$. Then the optimum $\pi^\star$ from Prop. 1 satisfies*

$$D_f\big(\pi^\star(\cdot \mid \boldsymbol{x}_1)\|(\tau_2^1 \sharp \pi^\star \sharp \tau_1^2)(\cdot \mid \boldsymbol{x}_1)^{\beta_1}\big) = 0 \quad and$$
$$D_f\big(\pi^\star(\cdot \mid \boldsymbol{x}_2)\|(\tau_1^2 \sharp \pi^\star \sharp \tau_2^1)(\cdot \mid \boldsymbol{x}_2)^{\beta_2}\big) = 0,$$

*i.e., $(f, 0)$-crosslingual consistency at $(\boldsymbol{x}_1, \boldsymbol{x}_2)$ in the sense of Def. 1.*

*Proof.* We prove the direction $L_1 \to L_2$; the symmetric direction follows by exchanging the roles of $L_1$ and $L_2$.

Invertibility of the translators says $\tau_1^2 \sharp \tau_2^1 = \mathbf{id}$ and $\tau_2^1 \sharp \tau_1^2 = \mathbf{id}$, so there is a bijection $\tau \colon L_1 \to L_2$ with $\tau_1^2(\boldsymbol{x}_2 \mid \boldsymbol{x}_1) = \mathbb{1}[\boldsymbol{x}_2 = \tau(\boldsymbol{x}_1)]$ and $\tau_2^1(\boldsymbol{y}_1 \mid \boldsymbol{y}_2) = \mathbb{1}[\boldsymbol{y}_1 = \tau_2^1(\boldsymbol{y}_2)]$. In particular, $p_1(\boldsymbol{y}_1 \mid \boldsymbol{x}_1) = (\tau_2^1 \sharp \rho \sharp \tau_1^2)(\boldsymbol{y}_1 \mid \boldsymbol{x}_1) = \rho(\tau_1^2(\boldsymbol{y}_1) \mid \tau_1^2(\boldsymbol{x}_1))$, and symmetrically $p_2(\boldsymbol{y}_2 \mid \boldsymbol{x}_2) = \rho(\tau_2^1(\boldsymbol{y}_2) \mid \tau_2^1(\boldsymbol{x}_2))$. By Prop. 1 and exact balance $\beta_1 \beta_2 = 1$,

$$\pi^\star(\boldsymbol{y}_1 \mid \boldsymbol{x}_1) \propto p_1(\boldsymbol{y}_1 \mid \boldsymbol{x}_1)^{\beta_1} \rho(\boldsymbol{y}_1 \mid \boldsymbol{x}_1) = \rho(\tau_1^2(\boldsymbol{y}_1) \mid \tau_1^2(\boldsymbol{x}_1))^{\beta_1} \rho(\boldsymbol{y}_1 \mid \boldsymbol{x}_1), \tag{19a}$$

$$\pi^\star(\boldsymbol{y}_2 \mid \boldsymbol{x}_2) \propto p_2(\boldsymbol{y}_2 \mid \boldsymbol{x}_2)^{\beta_2} \rho(\boldsymbol{y}_2 \mid \boldsymbol{x}_2) = \rho(\tau_2^1(\boldsymbol{y}_2) \mid \tau_2^1(\boldsymbol{x}_2))^{\beta_2} \rho(\boldsymbol{y}_2 \mid \boldsymbol{x}_2). \tag{19b}$$

Substituting $\boldsymbol{y}_2 = \tau_1^2(\boldsymbol{y}_1)$ and $\boldsymbol{x}_2 = \tau_1^2(\boldsymbol{x}_1)$ in the second line and raising to the power $\beta_1$,

$$\pi^\star(\tau_1^2(\boldsymbol{y}_1) \mid \tau_1^2(\boldsymbol{x}_1))^{\beta_1} \propto \rho(\boldsymbol{y}_1 \mid \boldsymbol{x}_1)^{\beta_1 \beta_2} \rho(\tau_1^2(\boldsymbol{y}_1) \mid \tau_1^2(\boldsymbol{x}_1))^{\beta_1} = \rho(\boldsymbol{y}_1 \mid \boldsymbol{x}_1) \rho(\tau_1^2(\boldsymbol{y}_1) \mid \tau_1^2(\boldsymbol{x}_1))^{\beta_1}, \tag{20}$$

using $\beta_1 \beta_2 = 1$ and $\tau_2^1 \sharp \tau_1^2 = \mathbf{id}$. The right-hand side coincides with the right-hand side of the first line of Eq. (19a) as a function of $\boldsymbol{y}_1$, so after normalization (both sides are probability distributions over $L_1$ given $\boldsymbol{x}_1$),

$$\pi^\star(\cdot \mid \boldsymbol{x}_1) = \pi^\star(\tau_1^2(\cdot) \mid \tau_1^2(\boldsymbol{x}_1))^{\beta_1} = (\tau_2^1 \sharp \pi^\star \sharp \tau_1^2)^{\beta_1}(\cdot \mid \boldsymbol{x}_1). \tag{21}$$

Therefore $D_f\big(\pi^\star(\cdot \mid \boldsymbol{x}_1)\|(\tau_2^1 \sharp \pi^\star \sharp \tau_1^2)^{\beta_1}(\cdot \mid \boldsymbol{x}_1)\big) = 0$. ∎

## C.3. Proof of Prop. 3

**Proposition 3.** *Assume $\mu_1$ and $\mu_2$ are supported on $L_1$ and $L_2$, respectively, with $L_1 \cap L_2 = \emptyset$. The minimizer $z^*$ of $\mathcal{D}$ (Eq. (15)), unique up to a prompt-dependent additive constant, is given by*

$$z^*(\boldsymbol{y} \mid \boldsymbol{x}) = \log \pi^\star(\boldsymbol{y} \mid \boldsymbol{x}), \tag{16}$$

*with $\pi^\star$ defined in Eq. (11).*

*Proof.* Due to the non-negativity of the norm, both $\mathcal{D}_1(\boldsymbol{\theta}, \boldsymbol{x}_1)$ and $\mathcal{D}_2(\boldsymbol{\theta}, \boldsymbol{x}_2)$ are non-negative, thus the minimum of $\mathcal{D}(\boldsymbol{\theta})$ is 0.

When $z_{\boldsymbol{\theta}}(\boldsymbol{y} \mid \boldsymbol{x}) = \log \pi^\star(\boldsymbol{y} \mid \boldsymbol{x})$, $\mathcal{D}(\boldsymbol{\theta})$ reaches its minimum.

When $\mathcal{D}(\boldsymbol{\theta}) = 0$, both $\mathcal{D}_1(\boldsymbol{\theta}, \boldsymbol{x}_1)$ and $\mathcal{D}_2(\boldsymbol{\theta}, \boldsymbol{x}_2)$ are 0, meaning that $z_{\boldsymbol{\theta}}(\boldsymbol{y} \mid \boldsymbol{x}) = \log \pi^\star(\boldsymbol{y} \mid \boldsymbol{x})$ for all $\boldsymbol{x}$ and $\boldsymbol{y}$. Thus, the minimizer $z^*$ of $\mathcal{D}$ (Eq. (15)) is $z^*(\boldsymbol{y} \mid \boldsymbol{x}) = \log \pi^\star(\boldsymbol{y} \mid \boldsymbol{x})$. ∎

# D. Generalization to $N$ Languages

We generalize §3 to $N$ languages, which we index by $m, n \in [N]$. Each language $L_m$ comes with a monolingual prompt distribution $\mu_m \in \Delta(L_m)$ and pairwise translators $\tau_m^n \colon L_m \rightsquigarrow L_n$ for $m \neq n$. Mirroring §3, define the round-trip target for prompts in $L_m$ routed via $L_n$:

$$p_{mn}(\cdot \mid \boldsymbol{x}) \stackrel{\text{def}}{=} \tau_n^m \sharp \rho \sharp \tau_m^n(\cdot \mid \boldsymbol{x}) \in \Delta(L_m), \qquad \boldsymbol{x} \in L_m, n \neq m. \tag{22}$$

Mirroring Eq. (9a), let $\mathbf{B} = [\beta_{mn}]_{m,n=1}^N \in \mathbb{R}_{>0}^{N \times N}$ be a positive strength matrix with $\beta_{mn}$ for $m \neq n$ scaling $\log p_{mn}$ and $\beta_{mm}$ scaling $\log \rho$ at $L_m$. The $N$-language objective decomposes into one language-specific half per language:

$$\mathcal{P}(\boldsymbol{\theta}) \stackrel{\text{def}}{=} \sum_{m=1}^N \mathbb{E}_{\boldsymbol{x} \sim \mu_m} \bigg[ \underbrace{\text{KL}\big(\pi_{\boldsymbol{\theta}}(\cdot \mid \boldsymbol{x})\|\rho(\cdot \mid \boldsymbol{x})\big)}_{\text{fidelity}} - \underbrace{\sum_{\substack{n=1, \\ n \neq m}}^N \beta_{mn} \log p_{mn}(\cdot \mid \boldsymbol{x})}_{\text{reward}} \bigg]. \tag{23}$$

The bilingual case ($N = 2$) recovers Eq. (9a) with $\beta_{11} = \beta_{22} = 1$, $\beta_{12} = \beta_1$, and $\beta_{21} = \beta_2$.

**Proposition 4.** *Let $\Sigma$ be an alphabet. Assume the languages $L_1, \ldots, L_N \subseteq \Sigma^*$ are pairwise disjoint. For each $m \in [N]$ and every $\boldsymbol{x} \in \mathrm{supp}(\mu_m)$, the unique minimizer of $\mathcal{P}$ in Eq. (23) is given by*

$$\pi^\star(\cdot \mid \boldsymbol{x}) \propto \rho(\cdot \mid \boldsymbol{x}) \prod_{\substack{n=1, \\ n \neq m}}^{N} p_{mn}(\cdot \mid \boldsymbol{x})^{\beta_{mn}}, \qquad \boldsymbol{x} \in L_m, \tag{24}$$

*with strength parameters independent of $\boldsymbol{x}$.*

*Proof.* The proof extends the bilingual case (Prop. 1). By the disjoint-support assumption, it suffices to consider $\boldsymbol{x} \in L_m$, in which case

$$\mathcal{P}(\boldsymbol{\theta}, \boldsymbol{x}) = \mathbb{E}_{\boldsymbol{y} \sim \pi_{\boldsymbol{\theta}}} \Big[ \log \pi_{\boldsymbol{\theta}}(\boldsymbol{y} \mid \boldsymbol{x}) - \sum_{\substack{n=1, \\ n \neq m}}^{N} \beta_{mn} \log(\tau_n^m \,\sharp\, \rho \,\sharp\, \tau_m^n)(\boldsymbol{y} \mid \boldsymbol{x}) \Big]$$

$$= \mathrm{KL}\left( \pi_{\boldsymbol{\theta}}(\cdot \mid \boldsymbol{x}) \Big\| \frac{\prod_{\substack{n=1, \\ n \neq m}}^{N}(\tau_n^m \,\sharp\, \rho \,\sharp\, \tau_m^n)(\cdot \mid \boldsymbol{x})^{\beta_{mn}} \rho(\cdot \mid \boldsymbol{x})}{Z_m(\boldsymbol{x})} \right) - \log Z_m(\boldsymbol{x}). \tag{25}$$

The minimizer of Eq. (25) is $\pi^\star(\cdot \mid \boldsymbol{x}) \propto \prod_{\substack{n=1, \\ n \neq m}}^{N} p_{mn}(\cdot \mid \boldsymbol{x})^{\beta_{mn}} \cdot \rho(\cdot \mid \boldsymbol{x})$ for $\boldsymbol{x} \in L_m$. ∎

Mirroring Prop. 2, we give a pairwise consistency guarantee when $\mathbf{B}$ is rank one.

**Proposition 5.** *Suppose matrix $\mathbf{B}$ is rank one, i.e., $\beta_{mn} = u_m v_n$ for some positive $\mathbf{u}, \mathbf{v} \in \mathbb{R}_{>0}^N$, as noted in §3.3, and $\beta_{mm} = 1$ for $m \in [N]$, and the pairwise translators $(\tau_m^n, \tau_n^m)$ for $m \neq n$ are invertible, i.e., $\tau_m^n \,\sharp\, \tau_n^m = \mathrm{id}$ and $\tau_n^m \,\sharp\, \tau_m^n = \mathrm{id}$. Moreover, the translators are mutually consistent: $\tau_m^\ell \,\sharp\, \tau_n^m = \tau_n^\ell$ for all distinct $m, n, \ell \in [N]$ (the cocycle identity), which holds in particular whenever all translators are restrictions of a single global bijection between the language sets. Suppose we have prompts in $N$ languages $(\boldsymbol{x}_1, \ldots, \boldsymbol{x}_N)$ that satisfy $\boldsymbol{x}_m = \tau_n^m(\boldsymbol{x}_n)$ for all $m \neq n$. Then, we have*

$$D_f\big(\pi^\star(\cdot \mid \boldsymbol{x}_m)^{u_n} \| (\tau_n^m \,\sharp\, \pi^\star \,\sharp\, \tau_m^n)(\cdot \mid \boldsymbol{x}_m)^{u_m}\big) = 0 \qquad \forall m \neq n. \tag{26}$$

*Proof.* First, note that $\beta_{mm} = u_m v_m = 1$ for $m \in [N]$. For any two languages $L_m, L_n$ where $m \neq n$,

$$\pi^\star(\boldsymbol{y}_m \mid \boldsymbol{x}_m) \propto \prod_{\ell \neq m,n} \rho(\tau_m^\ell(\boldsymbol{y}_m) \mid \tau_m^\ell(\boldsymbol{x}_m))^{u_m \mathbf{v}_\ell} \cdot \rho(\tau_m^n(\boldsymbol{y}_m) \mid \tau_m^n(\boldsymbol{x}_m))^{u_m v_n} \cdot \rho(\boldsymbol{y}_m \mid \boldsymbol{x}_m), \tag{27a}$$

$$\pi^\star(\boldsymbol{y}_n \mid \boldsymbol{x}_n) \propto \prod_{\ell \neq m,n} \rho(\tau_n^\ell(\boldsymbol{y}_n) \mid \tau_n^\ell(\boldsymbol{x}_n))^{u_n \mathbf{v}_\ell} \cdot \rho(\tau_n^m(\boldsymbol{y}_n) \mid \tau_n^m(\boldsymbol{x}_n))^{u_n v_m} \cdot \rho(\boldsymbol{y}_n \mid \boldsymbol{x}_n). \tag{27b}$$

Raising Eq. (27a) to the $u_n^{\mathrm{th}}$ power, Eq. (27b) to $u_m^{\mathrm{th}}$ power, we get

$$\pi^\star(\boldsymbol{y}_m \mid \boldsymbol{x}_m)^{u_n} = \pi^\star(\boldsymbol{y}_n \mid \boldsymbol{x}_n)^{u_m} = \prod_{\ell \neq n,m} \rho(\tau_m^\ell(\boldsymbol{y}_m) \mid \tau_m^\ell(\boldsymbol{x}_m))^{u_m u_n \mathbf{v}_\ell} \cdot \rho(\boldsymbol{y}_m \mid \boldsymbol{x}_m)^{u_n} \rho(\boldsymbol{y}_n \mid \boldsymbol{x}_n)^{u_m}, \tag{28}$$

using the fact that $u_n v_n = u_m v_m = 1$ and $\rho(\tau_m^\ell(\boldsymbol{y}_m) \mid \tau_m^\ell(\boldsymbol{x}_m)) = \rho(\tau_n^\ell(\boldsymbol{y}_n) \mid \tau_n^\ell(\boldsymbol{x}_n))$. The latter equality holds by the cocycle identity: since $\boldsymbol{y}_n = \tau_m^n(\boldsymbol{y}_m)$ and $\boldsymbol{x}_n = \tau_m^n(\boldsymbol{x}_m)$, we have $\tau_n^\ell(\boldsymbol{y}_n) = \tau_n^\ell(\tau_m^n(\boldsymbol{y}_m)) = \tau_m^\ell(\boldsymbol{y}_m)$, and likewise for $\boldsymbol{x}$, so the intermediate-language factors ($\ell \neq m, n$) coincide. Therefore, $D_f\big(\pi^\star(\cdot \mid \boldsymbol{x}_m)^{u_n} \| (\tau_n^m \,\sharp\, \pi^\star \,\sharp\, \tau_m^n)(\cdot \mid \boldsymbol{x}_m)^{u_m}\big) = 0$ for all $m \neq n$. ∎

| Dataset | Knowledge Type | #Langs | Parallel | | #Train | #Test | Response Format |
| --- | --- | --- | --- | --- | --- | --- | --- |
| | | | Prompt | Candidate | | | |
| MMMLU | General Knowledge | 12(+2) | ✓ | ✓ | 5000 | 9042 | A/B/C/D |
| XCSQA | Commonsense Reasoning | 16 | ✓ | ✓ | 800 | 200 | A/B/C/D/E |
| BMLAMA | Factual Association | 17 | ✓ | ✓ | 5000 | 1792 | Words |

*Table 6.* Statistics of MMMLU, XCSQA, and BMLAMA datasets.

| Dataset | Prompt | Candidates | Reference Completion |
| --- | --- | --- | --- |
| MMMLU | Which cells in the blood do not have a nucleus?
A. Lymphocyte
B. Monocyte
C. Erythrocyte
D. Basophil | [A, B, C, D] | C |
| XCSQA | What might lock after someone drives in?
A. gate
B. doorknob
C. mouths
D. entrance
E. front door | [A, B, C, D, E] | A |
| BMLAMA | Berlin is the capital of | [Greenland, Piedmont, Oman, Venezuela, Fiji, Latvia, Taiwan, Norway, Romania, Germany] | Germany |

*Table 7.* Examples of instances in MMMLU, XCSQA and BMLAMA.

## E. Implementation Details

In our experiments, we set the strength parameters $\beta_1 = \beta_2 = 1$ if not otherwise specified. For all models, we use the AdamW optimizer with a learning rate of $1\text{e}^{-5}$, with an exception of $1\text{e}^{-6}$ for Gemma models on XCSQA to avoid overfitting[6]. All models are trained on four A100 GPUs of 40GB memory. For SFT, DPO, and CALM, the learning rate is set to $5\text{e}^{-7}$, $5\text{e}^{-6}$, and $5\text{e}^{-6}$, respectively.

## F. Dataset Details

Three datasets with parallel prompts and candidate responses are used in our experiments. Here we list the detailed statistics in Table 6 and examples in Table 7.

**MMMLU.** MMMLU is a multilingual extension of the MMLU dataset (Hendrycks et al., 2021) on general knowledge. LMs are prompted with a factual query and candidate answers, and are expected to give a response from $\{A, B, C, D\}$. Each query and its candidate answers are given in 14 languages. In our experiments, we exclude Bengali due to the GPU constraint, and Swahili and Yoruba due to their low accuracy. We conduct extra experiments and in-depth analysis on Swahili and Yoruba in §5.4.

**XCSQA.** Similar to MMMLU, the prompts are multi-choice commonsense reasoning tasks over 16 languages. The response space is also a set of capital letters: $\{A, B, C, D, E\}$.

**BMLAMA.** The BMLAMA dataset (Qi et al., 2023) specifically focuses on factual associations, and the response format is objective words rather than option letters, which promotes the best crosslingual knowledge alignment in our experiments. Parallel prompts and candidate responses are provided across 17 languages.

---

[6]For the AdamW optimizer, we use a weight decay of 0, $\beta_1 = 0.9, \beta_2 = 0.999, \varepsilon = 1\text{e}^{-8}$.

**Sampling training instances.** Regarding MMMLU and BMLAMA, we randomly sample two pairs of parallel candidate responses per prompt in the training set, yielding 5000 instances for DCO. Regarding XCSQA, we repeat this sampling procedure seven times to construct a dataset of comparable size ($800 \cdot 7 = 5600$) for DCO training.

## G. Definition of RankC: Ranking-based Crosslingual Consistency

RankC (Qi et al., 2023) is a ranking-based consistency metric for assessing crosslingual knowledge consistency independently of probing accuracy. Given a pair of prompts $x_1, x_2$ that satisfy $x_1 = \tau_2^1(x_2)$, assuming there are $M$ candidate responses $\{c^j\}_{j=1}^M$, we sort the candidate responses in each language by descending likelihood into $c_1^1, c_1^2, \ldots, c_1^M \in L_1$ and $c_2^1, c_2^2, \ldots, c_2^M \in L_2$. First, the 'precision at $j$' (denoted P@$j$) is defined as the proportion of overlap among the top-$j$ candidates in both languages: P@$j = \frac{1}{j} |\{c_1^1, \ldots, c_1^j\} \cap \{c_2^1, \ldots, c_2^j\}|$. An exponentially decaying weight $w_j = \frac{\exp(M-j)}{\sum_{k=1}^M \exp(M-k)}$ is multiplied to each P@$j$, so that agreements at smaller $j$ (i.e., high-likelihood candidate responses) are rewarded more. RankC is defined as

$$\text{RankC}(x_1, x_2) = \sum_{j=1}^M w_j \cdot \text{P@}j. \tag{29}$$

## H. Direction Controlling Results on English–Yoruba

The original accuracy on YO is worse than SW. The **default** strength parameters induce decreases in both languages: EN drops by $-15.94$ and YO also declines by $-1.18$. Making Yoruba 'stable' (YO-anchored; $\beta_1=10$, $\beta_2=0.1$) exacerbates the problem, especially pushing EN down to $-19.97$ while still not helping Yoruba accuracy ($\Delta = -1.22$). In contrast, the EN-anchored setup ($\beta_1=0.1$, $\beta_2=10$) delivers the best trade-off: EN accuracy remains less affected ($+2.98$) while that of Yoruba also improves by $+1.43$. Regarding CLC, for EN–YO, the baseline of 45.67 increases to 57.16, 55.40, and 51.66, demonstrating the effectiveness of DCO in crosslingual knowledge alignment. As for the proportion of changed responses, similar to EN-SW, EN-anchored minimizes EN updates as 19.00% while allowing substantial revisions on Yoruba as 59.87%. Default setup raises EN changes to 38.74%, and YO-anchored setup further increases EN changes to 42.51%.

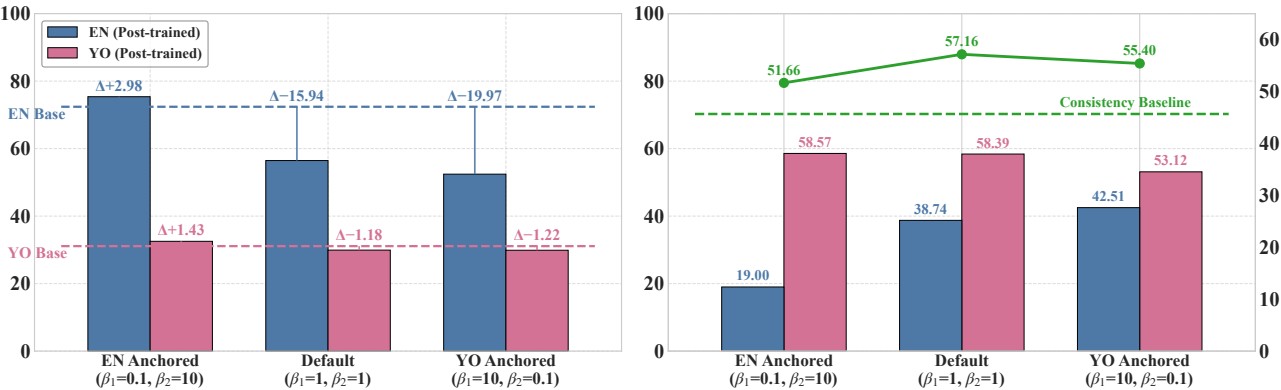

*Figure 4.* Left: Response accuracy after performing DCO on English–Yoruba. Right: Proportion of prompts for which the LM's response changes after DCO, with CLC values marked in green.

## I. Guidance of Direction Controlling for Real-World Applications

Direction-controlling strength parameters matter: by adjusting $(\beta_1, \beta_2)$, we control the transfer strength for each side, thus deciding the direction of optimizing knowledge consistency. The low-resource languages Swahili and Yoruba were selected in our experiments for the best visibility of the effect; yet the results should not be misinterpreted as meaning 'always set a large strength parameter $\beta$ for EN.' In fact, the principle is more general: anchor on the high-quality, high-priority language, which is EN in our study, but could be French or any other well-trained language, according to the specific requirements of the downstream LM application. When both languages are of comparable quality, or when policy requires reciprocity, a symmetric schedule like 1:1 is expected to be optimal. In practice, $(\beta_1, \beta_2)$ can also be selected empirically against a small validation set. The 'ratio of changed responses' (i.e., Fig. 3 (right) & Fig. 4 (right)) is useful: if the intended stable language

| Method | Qwen2.5-14B | | | Gemma3-12B-pt | | | Qwen3-14B | | | Aya-Expanse-8B | | | Llama3.1-8B | | |
|---|---|---|---|---|---|---|---|---|---|---|---|---|---|---|---|
| | CLC$_{all}$ | A$_{EN}$ | A$_{\neg EN}$ | CLC$_{all}$ | A$_{EN}$ | A$_{\neg EN}$ | CLC$_{all}$ | A$_{EN}$ | A$_{\neg EN}$ | CLC$_{all}$ | A$_{EN}$ | A$_{\neg EN}$ | CLC$_{all}$ | A$_{EN}$ | A$_{\neg EN}$ |
| Base | 68.6±0.02 | 72.5±0.4 | 58.1±0.2 | 73.6±0.02 | 70.1±0.4 | 62.3±0.2 | 76.1±0.02 | 76.6±0.4 | 67.3±0.1 | 72.2±0.02 | 59.8±0.5 | 52.9±0.2 | 60.9±0.02 | 57.3±0.5 | 45.8±0.2 |
| + SFT* | +0.6±0.02 | +1.5±0.4 | +6.7±0.2 | +0.6±0.02 | +0.7±0.4 | +1.6±0.2 | −0.2±0.02 | +0.1±0.5 | +0.5±0.1 | +3.5±0.02 | +0.7±0.5 | +0.5±0.2 | +4.3±0.02 | +6.7±0.5 | +5.9±0.2 |
| + DPO* | +12.3±0.02 | +7.8±0.4 | +13.9±0.1 | +6.5±0.02 | +1.8±0.4 | +3.4±0.1 | +3.0±0.02 | +2.7±0.5 | +4.2±0.1 | +1.3±0.02 | +2.5±0.5 | +2.5±0.2 | +10.1±0.02 | +8.0±0.4 | +8.8±0.2 |
| + DCO* | +13.1±0.02 | +7.6±0.4 | +13.5±0.1 | +10.2±0.02 | +1.2±0.4 | +2.9±0.1 | +4.4±0.02 | +2.8±0.5 | +4.3±0.1 | +3.1±0.02 | +2.7±0.5 | +2.6±0.2 | +13.8±0.02 | +7.3±0.4 | +8.9±0.2 |
| + CALM | +4.2±0.02 | +0.0±0.4 | +4.1±0.2 | +3.0±0.02 | −0.4±0.4 | −0.0±0.2 | +0.3±0.02 | −2.1±0.5 | −1.1±0.1 | +1.4±0.02 | −2.2±0.5 | −2.1±0.2 | +3.0±0.02 | −2.2±0.5 | −5.0±0.2 |
| + DCO | +10.6±0.02 | +4.0±0.4 | +9.6±0.2 | +6.5±0.02 | +0.9±0.4 | +2.5±0.2 | +2.7±0.02 | +0.4±0.5 | +1.3±0.1 | +5.3±0.02 | +0.5±0.5 | +0.5±0.2 | +9.4±0.02 | +7.5±0.5 | +7.6±0.2 |

*Table 8.* Comparison with previous methods in the joint training setup, on the MMMLU dataset. CLC$_{all}$: average crosslingual consistency (measured by RankC) between all language pairs; A$_{EN}$/A$_{\neg EN}$: average accuracy on English/non-English instances. We report mean scores with their standard errors. Methods with * are trained on reference completions.

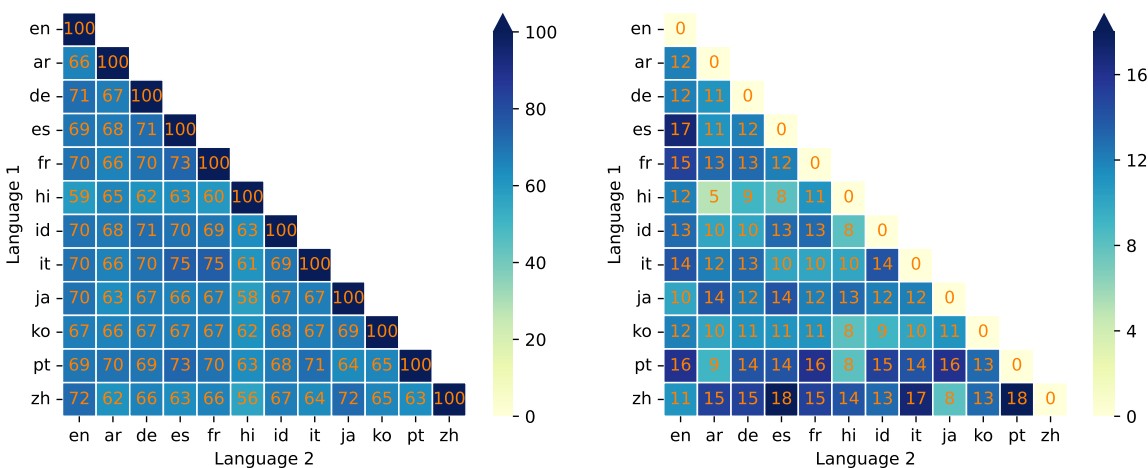

*Figure 5.* The changes in CLC of `Qwen2.5-14B` after DCO. Left: CLC between all language pairs on the original model. Right: The improvements in CLC of the post-DCO model.

($L_1$) exhibits excessive changes, or the language expected to shift shows little movement, reduce $\beta_1$, and by construction, $\beta_2$ will increase correspondingly ($\beta_2 = 1/\beta_1$).

Note that DCO is capable of improving CLC under any direction parameter setups, as demonstrated in Fig. 3 & Fig. 4. This empirical adjustment further yields gains in accuracy for both sides.

## J. Standard Deviation and Detailed CLC Results in the Multilingual Experiments

We present the standard deviation of the joint training experiment in App. J. We visualize the improvement of CLC between all language pairs in Fig. 5 to Fig. 9. The left sub-figure in each panel reports the baseline CLC scores, while the right sub-figure shows the absolute change after applying DCO. Warmer colors indicate higher CLC scores, and the delta plots highlight systematic gains across most language pairs. Notably, DCO consistently improves CLC, with substantial gains not only between typologically similar languages such as English–Spanish, but also between distant ones. For instance, Arabic–Chinese and Hindi–Japanese improve by 15% and 13%, respectively, while the Korean–French pair gains a remarkable 14% increase on Llama-3.1-8B. These results indicate that DCO not only raises overall knowledge consistency but also narrows the gap between distant language families.

## K. Full Bilingual Experimental Results

We present the detailed results of the bilingual experiments in Tables 10 to 15

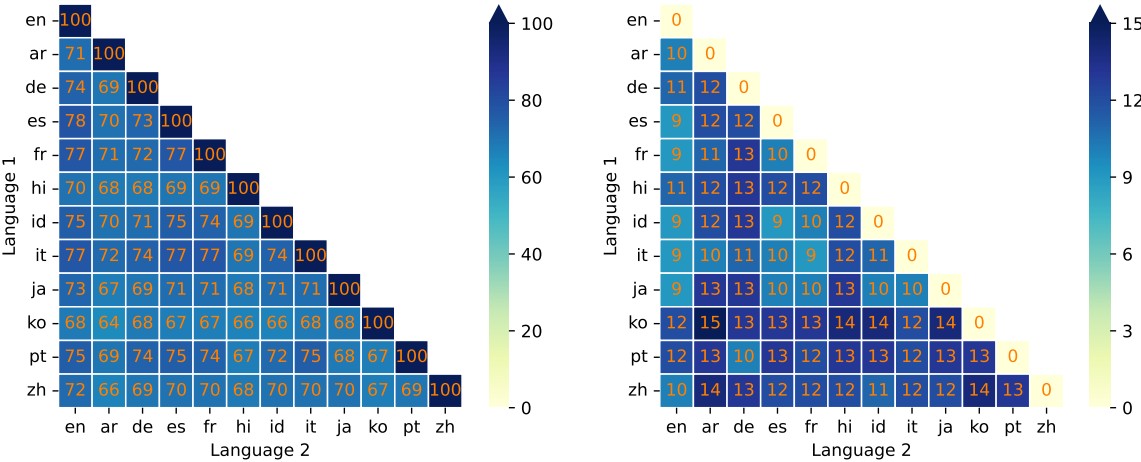

Figure 6. The changes in CLC of Gemma3-12B after DCO. Left: CLC between all language pairs on the original model. Right: The Improvements in CLC of the post-DCO model.

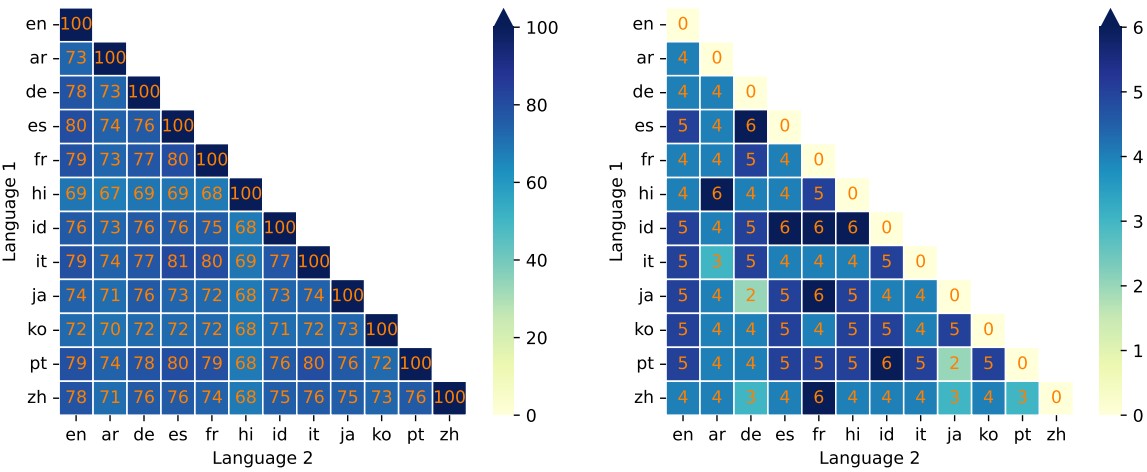

Figure 7. The changes in CLC of Qwen3-14B after DCO. Left: CLC between all language pairs on the original model. Right: The Improvements in CLC of the post-DCO model.

## L. Bilingual Knowledge Alignment Beyond English

While our main experiments instantiate a bilingual DCO that aligns English with a single non-English language (motivated by English having the highest probing accuracy in our preliminary study), it is important to verify that the effect is not specific to using English as an anchor. Therefore, we evaluate DCO on Qwen2.5-14B with non-English languages in BMLAMA, covering both distant and less-distant pairs: (i) Arabic–Chinese (AR–ZH) (typologically distant), (ii) Korean–French (KO–FR) (typologically distant), (iii) English–Spanish (EN–ES) (closer pair for reference).

The results in Table 9 show that DCO improves CLC and accuracy across all tested language pairs, including typologically distant ones. In particular, CLC increases by +14.35% (EN–ES), +10.40% (AR–ZH), and +14.95% (KO–FR), while both languages' accuracies rise in every pair. Although distant pairs start from lower baseline consistency (expected due to translation/linguistic gaps), DCO still delivers substantial gains, especially between Korean and French, suggesting its effectiveness is not dependent on typological similarity.

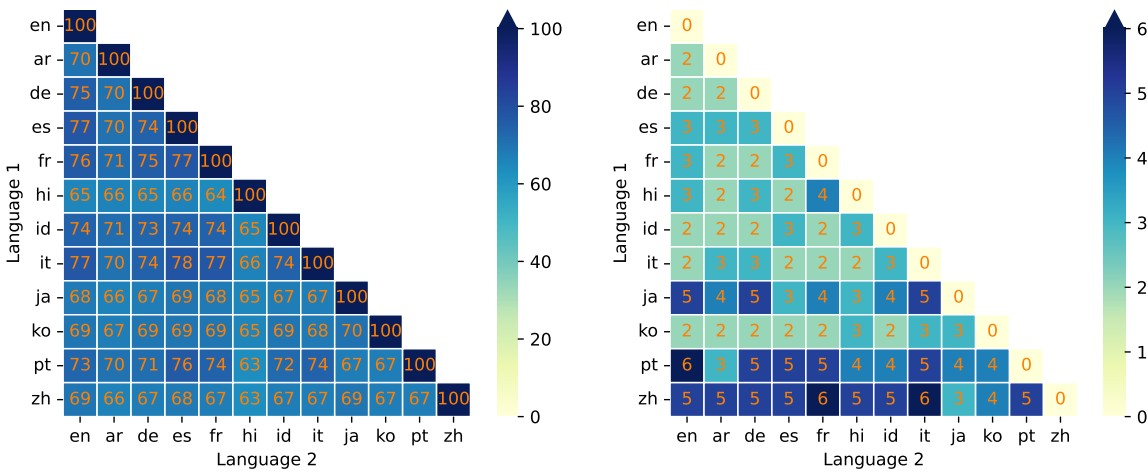

*Figure 8.* The changes in CLC of `Aya-Expanse-8B` after DCO. Left: CLC between all language pairs on the original model. Right: The Improvements in CLC of the post-DCO model.

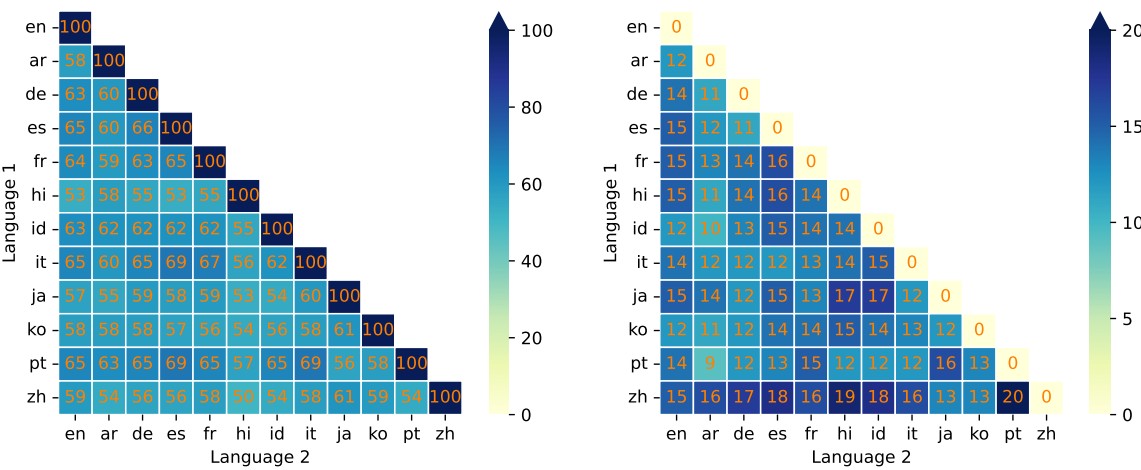

*Figure 9.* The changes in CLC of `Llama3.1-8B` after DCO. Left: CLC between all language pairs on the original model. Right: The Improvements in CLC of the post-DCO model.

| Model | EN-ES | | | AR-ZH | | | KO-FR | | |
|---|---|---|---|---|---|---|---|---|---|
| | Acc EN | Acc ES | CLC | Acc AR | Acc ZH | CLC | Acc KO | Acc FR | CLC |
| Qwen2.5-14B | 62.67 | 44.81 | 47.45 | 42.35 | 37.05 | 36.34 | 40.57 | 36.71 | 37.29 |
| +DCO | **68.97** | **57.37** | **61.80** | **50.33** | **48.49** | **46.74** | **52.46** | **53.68** | **52.24** |

*Table 9.* Non-English alignment results on BMLAMA.

## M. Full Results of Out-of-Domain Experiments

We present the full results on all five out-of-domain tasks in Table 16 and Table 17. The improvement in both evaluation dimensions suggests the significant generalizability of DCO.

|  | AR | DE | ES | FR | HI | ID | IT | JA | KO | PT | ZH |
|---|---|---|---|---|---|---|---|---|---|---|---|
| **Consistency** | | | | | | | | | | | |
| Qwen2.5-7B | 64.18 | 70.68 | 74.60 | 73.33 | 50.95 | 69.20 | 73.46 | 68.05 | 65.63 | 72.40 | 72.38 |
| + DCO | +9.17 | +9.20 | +6.68 | +8.88 | +14.66 | +7.42 | +7.50 | +8.16 | +8.01 | +9.99 | +7.66 |
| Qwen2.5-14B | 66.19 | 70.93 | 69.45 | 70.19 | 59.15 | 70.43 | 69.68 | 70.16 | 67.02 | 68.54 | 72.38 |
| +DCO | +12.09 | +12.82 | +14.68 | +14.11 | +12.98 | +11.04 | +15.66 | +9.15 | +10.80 | +15.76 | +9.54 |
| Gemma3-4B-pt | 63.21 | 68.24 | 71.78 | 71.32 | 56.84 | 64.48 | 70.53 | 58.28 | 60.73 | 69.00 | 60.75 |
| +DCO | +10.69 | +10.07 | +9.40 | +8.11 | +18.41 | +11.22 | +8.73 | +14.09 | +11.32 | +4.85 | +14.08 |
| Gemma3-12B-pt | 70.56 | 73.89 | 77.93 | 76.81 | 69.82 | 75.23 | 76.87 | 72.96 | 68.13 | 74.86 | 72.11 |
| +DCO | +7.37 | +8.44 | +4.64 | +4.58 | +9.07 | +8.41 | +6.11 | +6.79 | +9.91 | +7.93 | +5.45 |
| Qwen3-8B | 65.17 | 71.58 | 75.31 | 72.37 | 65.16 | 68.52 | 74.85 | 71.64 | 67.00 | 74.85 | 76.37 |
| +DCO | +9.48 | +9.77 | +7.17 | +9.10 | +4.80 | +10.61 | +7.43 | +5.80 | +7.68 | +7.73 | +2.99 |
| Qwen3-14B | 72.70 | 77.84 | 80.34 | 78.90 | 68.56 | 76.17 | 79.38 | 74.06 | 72.46 | 79.30 | 77.72 |
| +DCO | +3.73 | +4.83 | +4.50 | +4.86 | +4.08 | +5.29 | +4.66 | +6.03 | +4.89 | +5.10 | +4.75 |
| Aya-Expanse-8B | 69.66 | 74.58 | 77.42 | 76.26 | 65.44 | 73.74 | 76.70 | 68.45 | 68.69 | 73.49 | 69.39 |
| +DCO | +5.37 | +4.70 | +4.01 | +5.17 | +5.23 | +4.17 | +4.86 | +7.41 | +5.04 | +6.59 | +6.06 |
| Llama3.1-8B | 58.31 | 62.70 | 65.23 | 63.83 | 52.67 | 62.54 | 65.38 | 56.91 | 57.59 | 65.15 | 59.31 |
| +DCO | +10.75 | +10.94 | +12.57 | +13.89 | +14.88 | +7.99 | +6.21 | +14.50 | +11.07 | +13.73 | +16.14 |
| Llama3.2-3B | 46.07 | 54.38 | 54.06 | 53.65 | 42.78 | 51.77 | 53.25 | 45.11 | 42.75 | 56.23 | 47.82 |
| + DCO | +16.54 | +12.60 | +16.36 | +17.28 | +18.01 | +16.39 | +16.53 | +17.38 | +17.27 | +17.03 | +18.97 |

*Table 10.* Consistency improvements with English for each language across all models on MMMLU.

|  | EN | AR | DE | ES | FR | HI | ID | IT | JA | KO | PT | ZH |
|---|---|---|---|---|---|---|---|---|---|---|---|---|
| **Accuracy** | | | | | | | | | | | | |
| Qwen2.5-7B | 69.57 | 54.51 | 59.98 | 63.43 | 62.30 | 44.33 | 58.71 | 62.66 | 59.07 | 57.13 | 61.93 | 63.98 |
| + DCO | -0.01 | +1.24 | +2.02 | +0.72 | +1.46 | +4.32 | +1.80 | +1.76 | +1.74 | +1.44 | +1.97 | +1.50 |
| Qwen2.5-14B | 72.46 | 54.88 | 59.52 | 56.83 | 58.31 | 45.66 | 61.06 | 56.21 | 63.24 | 56.96 | 56.35 | 69.87 |
| +DCO | +1.64 | +7.63 | +8.51 | +12.94 | +10.67 | +8.41 | +6.31 | +12.57 | +4.16 | +8.00 | +13.38 | +0.83 |
| Gemma3-4B-pt | 56.09 | 43.14 | 49.02 | 52.51 | 50.51 | 39.64 | 48.47 | 49.64 | 42.78 | 45.39 | 50.25 | 46.56 |
| +DCO | +0.08 | +1.04 | +2.77 | -0.02 | +0.26 | +5.32 | +2.93 | +1.60 | +4.64 | +2.13 | +1.81 | +3.07 |
| Gemma3-12B-pt | 70.07 | 58.60 | 64.19 | 65.43 | 64.14 | 57.94 | 62.07 | 65.05 | 60.83 | 60.49 | 65.01 | 61.36 |
| +DCO | -0.90 | +1.71 | +1.22 | -0.70 | +0.73 | +4.44 | +1.39 | +0.54 | +1.46 | +1.51 | +1.28 | +1.33 |
| Qwen3-8B | 71.15 | 56.34 | 63.41 | 65.60 | 64.58 | 54.29 | 60.99 | 64.78 | 61.32 | 59.47 | 64.11 | 66.29 |
| +DCO | +0.72 | +3.03 | +2.69 | +2.08 | +1.84 | +1.66 | +2.89 | +1.94 | +1.42 | +1.74 | +2.87 | +0.31 |
| Qwen3-14B | 76.58 | 63.07 | 68.56 | 71.04 | 70.10 | 58.69 | 67.49 | 70.35 | 65.65 | 64.74 | 70.72 | 69.64 |
| +DCO | +0.13 | +0.80 | +2.36 | +1.38 | +1.41 | +1.98 | +2.09 | +1.50 | +2.10 | +1.52 | +1.68 | +1.55 |
| Aya-Expanse-8B | 59.76 | 50.75 | 54.10 | 56.36 | 55.71 | 46.20 | 53.14 | 54.62 | 51.69 | 51.32 | 55.63 | 52.57 |
| +DCO | +0.52 | +0.46 | -0.01 | -0.28 | -0.03 | +0.96 | +0.57 | +0.66 | +1.37 | +0.01 | +0.72 | +0.68 |
| Llama3.1-8B | 57.27 | 41.19 | 49.13 | 51.17 | 47.56 | 35.01 | 46.88 | 49.89 | 44.63 | 43.10 | 48.71 | 46.40 |
| +DCO | +0.74 | +3.33 | +3.21 | +4.16 | +5.07 | +0.23 | +3.39 | +0.32 | +2.12 | +2.91 | +4.82 | +3.54 |
| Llama3.2-3B | 52.23 | 35.45 | 43.86 | 46.25 | 44.67 | 33.92 | 42.97 | 43.80 | 34.62 | 36.16 | 45.07 | 38.41 |
| + DCO | +0.93 | +4.64 | +2.48 | +1.75 | +2.91 | +3.17 | +1.66 | +2.83 | +5.96 | +4.42 | +2.16 | +4.90 |

*Table 11.* Accuracy of each model on MMMLU across all languages after DCO.

| | ZH | DE | ES | FR | IT | JA | NL | PL | PT | RU | AR | VI | HI | SW | UR |
|---|---|---|---|---|---|---|---|---|---|---|---|---|---|---|---|
| | | | | | | | **Consistency** | | | | | | | | |
| Qwen2.5-7B | 64.41 | 72.26 | 77.82 | 71.57 | 71.96 | 57.78 | 65.49 | 65.41 | 75.27 | 66.67 | 65.49 | 68.80 | 49.76 | 35.58 | 48.02 |
| + DCO | +10.81 | +4.33 | +4.50 | +6.55 | +6.04 | +8.78 | +9.73 | +6.33 | +5.97 | +6.34 | +8.33 | +6.32 | +8.25 | -0.28 | +7.83 |
| Qwen2.5-14B | 66.94 | 70.90 | 73.82 | 71.60 | 73.58 | 64.55 | 68.82 | 65.03 | 74.13 | 64.31 | 64.86 | 64.41 | 53.68 | 39.50 | 52.57 |
| + DCO | +8.03 | +5.80 | +8.21 | +5.93 | +3.81 | +5.92 | +8.52 | +8.34 | +6.52 | +8.73 | +11.00 | +10.68 | +5.33 | +0.40 | +4.90 |
| Gemma3-4B-pt | 54.23 | 60.91 | 57.85 | 56.83 | 60.28 | 50.21 | 56.93 | 53.29 | 57.68 | 57.18 | 53.50 | 52.78 | 51.56 | 37.36 | 47.26 |
| + DCO | +4.76 | +3.34 | +2.91 | +3.56 | +4.05 | +7.43 | +5.72 | +4.87 | +4.18 | +4.31 | +6.91 | +6.42 | +6.95 | +9.48 | +10.98 |
| Gemma3-12B-pt | 64.61 | 65.09 | 64.18 | 61.80 | 63.32 | 55.11 | 62.31 | 56.16 | 65.07 | 58.16 | 50.77 | 56.15 | 54.15 | 46.19 | 51.06 |
| + DCO | +3.40 | +3.27 | +6.15 | +4.54 | +4.88 | +3.51 | +4.99 | +5.94 | +1.54 | +5.76 | +7.04 | +5.72 | +3.35 | +0.65 | +8.45 |
| Qwen3-8B | 64.88 | 65.84 | 67.75 | 68.77 | 70.95 | 58.49 | 64.60 | 63.01 | 69.41 | 60.12 | 61.89 | 65.46 | 55.63 | 37.42 | 49.25 |
| + DCO | +4.42 | +6.58 | +8.88 | +5.64 | +4.60 | +4.19 | +7.06 | +6.84 | +6.63 | +7.25 | +10.64 | +5.01 | +4.52 | +5.19 | +5.69 |
| Qwen3-14B | 66.18 | 66.33 | 66.11 | 67.36 | 70.93 | 60.14 | 63.83 | 61.64 | 67.19 | 65.90 | 64.08 | 64.07 | 55.00 | 37.59 | 52.26 |
| + DCO | +4.36 | +8.36 | +12.06 | +7.23 | +3.29 | +6.19 | +9.93 | +8.55 | +8.70 | +4.54 | +5.40 | +7.32 | +6.53 | +4.88 | +9.70 |
| Aya-Expanse-8B | 66.36 | 69.33 | 72.63 | 71.18 | 69.02 | 57.02 | 67.24 | 64.38 | 65.18 | 64.96 | 65.29 | 65.77 | 59.01 | 35.93 | 45.34 |
| + DCO | +4.79 | +4.90 | +6.43 | +5.29 | +6.59 | +6.92 | +7.01 | +8.14 | +11.86 | +3.99 | +6.21 | +4.15 | +6.83 | +4.19 | +8.07 |
| Llama3.1-8B | 59.24 | 67.62 | 67.08 | 67.90 | 65.36 | 56.66 | 65.25 | 63.13 | 68.35 | 60.26 | 58.52 | 64.64 | 55.40 | 36.92 | 47.12 |
| + DCO | +11.29 | +3.89 | +12.32 | +5.15 | +8.13 | +6.75 | +5.51 | +5.56 | +4.56 | +9.53 | +6.21 | +2.45 | +14.60 | +24.20 | +16.28 |
| Llama3.2-3B | 57.52 | 64.49 | 65.19 | 62.84 | 61.03 | 53.57 | 59.29 | 54.31 | 61.62 | 60.54 | 59.98 | 57.48 | 52.58 | 35.28 | 45.53 |
| + DCO | +10.30 | +6.09 | +8.24 | +10.39 | +8.24 | +8.28 | +3.54 | +9.87 | +9.23 | +4.94 | -1.01 | +10.97 | +7.92 | +5.51 | +8.68 |

*Table 12.* Consistency improvements with English for each language across all models on XCSQA.

| | EN | ZH | DE | ES | FR | IT | JA | NL | PL | PT | RU | AR | VI | HI | SW | UR |
|---|---|---|---|---|---|---|---|---|---|---|---|---|---|---|---|---|
| | | | | | | | | **Accuracy** | | | | | | | | |
| Qwen2.5-7B | 84.00 | 55.50 | 58.00 | 66.00 | 60.50 | 62.00 | 52.50 | 54.50 | 57.50 | 63.00 | 55.50 | 53.50 | 61.50 | 39.50 | 25.00 | 41.00 |
| + DCO | -1.93 | +7.50 | +5.00 | +1.50 | +2.50 | +5.00 | +4.50 | +5.00 | +6.50 | +4.50 | +3.00 | +5.00 | +1.50 | +7.00 | 0.00 | +4.00 |
| Qwen2.5-14B | 87.00 | 59.50 | 62.00 | 65.00 | 62.50 | 65.00 | 57.50 | 61.00 | 60.00 | 67.50 | 56.00 | 54.50 | 56.50 | 46.50 | 32.50 | 47.00 |
| + DCO | -2.53 | +7.00 | +5.00 | +5.00 | +6.00 | +2.00 | +5.00 | +5.00 | +3.00 | +4.00 | +5.00 | +8.00 | +10.00 | +1.50 | 0.00 | +3.50 |
| Gemma3-4B-pt | 56.50 | 35.50 | 41.00 | 45.00 | 45.00 | 42.50 | 32.00 | 40.00 | 39.00 | 34.50 | 39.00 | 36.50 | 43.00 | 30.50 | 27.00 | 31.00 |
| + DCO | +2.30 | +5.50 | +3.50 | +0.50 | +0.50 | +4.00 | +3.50 | +4.00 | +1.00 | +3.50 | -0.50 | +2.50 | 0.00 | +2.50 | +1.50 | +3.50 |
| Gemma3-12B-pt | 66.00 | 52.50 | 52.50 | 53.50 | 52.00 | 56.50 | 43.00 | 53.00 | 45.50 | 53.50 | 43.00 | 39.50 | 46.50 | 40.00 | 37.50 | 40.00 |
| + DCO | +0.10 | +4.00 | +3.50 | +2.50 | +4.00 | +1.00 | +1.50 | +1.50 | +5.00 | +2.50 | +6.00 | +6.00 | +5.00 | +2.00 | +1.50 | +7.50 |
| Qwen3-8B | 73.50 | 53.00 | 53.50 | 56.50 | 61.00 | 60.50 | 49.00 | 55.50 | 53.50 | 57.00 | 51.50 | 50.50 | 56.50 | 47.00 | 28.00 | 41.50 |
| + DCO | +0.97 | +4.50 | +2.50 | +3.50 | -1.00 | +0.50 | -1.00 | +1.50 | +3.00 | +5.50 | +2.50 | +5.50 | +1.00 | +2.50 | +1.00 | +3.50 |
| Qwen3-14B | 77.50 | 55.00 | 58.00 | 60.00 | 57.00 | 62.50 | 51.50 | 59.00 | 57.00 | 59.00 | 57.00 | 55.50 | 60.50 | 47.50 | 27.50 | 43.00 |
| + DCO | +1.07 | +3.50 | +4.00 | +5.00 | +5.00 | +2.50 | +4.50 | +2.50 | +5.00 | +4.00 | +1.00 | +1.50 | +2.50 | +1.00 | +5.50 | +9.00 |
| Aya-Expanse-8B | 78.00 | 61.00 | 59.00 | 64.00 | 58.00 | 62.00 | 49.50 | 59.50 | 57.00 | 60.00 | 58.50 | 56.50 | 59.50 | 50.50 | 25.00 | 36.00 |
| + DCO | +0.57 | +4.00 | +2.50 | +5.00 | +4.50 | +3.00 | +5.50 | +1.00 | +4.00 | +7.00 | -2.00 | +2.50 | +4.00 | +3.50 | +3.50 | +7.00 |
| Llama3.1-8B | 67.50 | 48.00 | 55.50 | 54.50 | 55.00 | 56.50 | 41.00 | 52.00 | 51.00 | 53.00 | 48.50 | 47.00 | 55.50 | 39.50 | 27.00 | 32.00 |
| + DCO | +0.17 | +3.50 | 0.00 | +0.50 | +0.50 | +3.00 | -0.50 | +3.00 | +1.00 | -1.50 | +2.00 | -1.50 | +0.50 | +5.00 | +2.00 | +3.50 |
| Llama3.2-3B | 65.00 | 51.00 | 47.50 | 46.50 | 42.50 | 45.50 | 45.00 | 39.50 | 41.50 | 45.50 | 44.00 | 45.00 | 43.00 | 35.00 | 29.00 | 34.50 |
| + DCO | -4.07 | -3.50 | +3.00 | +8.50 | +6.00 | +3.50 | +4.00 | +2.50 | +8.50 | +3.50 | +2.00 | -1.00 | +8.00 | +4.50 | +1.50 | +4.00 |

*Table 13.* Accuracy of each model on XCSQA across all languages after DCO.

| | FR | NL | ES | RU | JA | ZH | KO | VI | EL | HU | HE | TR | CA | AR | UK | FA |
|---|---|---|---|---|---|---|---|---|---|---|---|---|---|---|---|---|
| **Consistency** | | | | | | | | | | | | | | | | |
| Qwen2.5-7B | 44.12 | 48.41 | 45.50 | 41.78 | 39.05 | 40.18 | 36.50 | 44.54 | 30.88 | 30.66 | 30.88 | 34.29 | 39.16 | 40.89 | 40.53 | 35.85 |
| + DCO | +17.90 | +13.98 | +16.21 | +15.08 | +16.89 | +11.53 | +18.40 | +19.05 | +17.79 | +15.54 | +17.27 | +17.10 | +14.64 | +13.24 | +15.02 | +14.15 |
| Qwen2.5-14B | 45.01 | 50.87 | 47.45 | 42.83 | 41.94 | 42.61 | 41.19 | 46.91 | 35.70 | 32.04 | 38.84 | 37.99 | 41.71 | 41.78 | 42.48 | 40.58 |
| + DCO | +17.33 | +14.48 | +14.35 | +16.05 | +16.66 | +9.77 | +16.23 | +16.32 | +18.86 | +16.48 | +14.32 | +15.93 | +13.36 | +15.40 | +16.67 | +14.38 |
| Gemma3-4B-pt | 38.26 | 48.78 | 45.63 | 40.57 | 36.99 | 31.78 | 35.72 | 45.23 | 38.16 | 34.85 | 35.91 | 39.77 | 36.88 | 35.51 | 43.75 | 35.92 |
| + DCO | +23.36 | +16.47 | +15.47 | +18.24 | +15.90 | +14.32 | +16.87 | +21.56 | +19.32 | +20.07 | +14.65 | +19.58 | +18.49 | +12.63 | +15.81 | +12.18 |
| Gemma3-12B-pt | 41.81 | 51.38 | 47.80 | 42.90 | 40.98 | 34.76 | 40.39 | 47.82 | 41.29 | 39.53 | 40.16 | 42.17 | 39.82 | 38.87 | 45.96 | 40.10 |
| + DCO | +23.27 | +15.53 | +15.53 | +17.37 | +15.17 | +15.12 | +15.74 | +20.81 | +19.02 | +18.83 | +13.72 | +19.46 | +17.92 | +11.74 | +15.76 | +11.35 |
| Qwen3-8B | 43.21 | 47.24 | 43.21 | 38.43 | 33.25 | 37.43 | 33.89 | 45.29 | 32.92 | 30.50 | 30.26 | 35.43 | 38.58 | 38.86 | 40.70 | 34.29 |
| + DCO | +15.21 | +14.19 | +13.47 | +14.05 | +16.92 | +11.13 | +16.15 | +13.71 | +14.91 | +16.06 | +12.46 | +14.27 | +13.44 | +11.47 | +14.45 | +13.91 |
| Qwen3-14B | 42.45 | 48.82 | 44.37 | 37.93 | 35.68 | 38.56 | 36.01 | 43.17 | 36.10 | 31.49 | 33.53 | 37.95 | 40.46 | 39.62 | 40.71 | 35.62 |
| + DCO | +16.43 | +14.11 | +15.01 | +17.66 | +20.46 | +12.22 | +16.78 | +17.53 | +15.39 | +18.40 | +14.80 | +14.37 | +15.12 | +13.71 | +18.13 | +18.09 |
| Aya-Expanse-8B | 50.81 | 51.38 | 58.03 | 39.84 | 39.72 | 38.32 | 36.77 | 55.08 | 34.74 | 29.12 | 36.97 | 41.80 | 37.02 | 40.11 | 44.69 | 36.43 |
| + DCO | +18.33 | +13.44 | +13.16 | +15.42 | +11.08 | +10.74 | +11.72 | +14.93 | +12.77 | +7.25 | +9.40 | +13.45 | +9.82 | +11.05 | +12.31 | +11.69 |
| Llama3.1-8B | 44.16 | 51.07 | 47.40 | 41.13 | 41.35 | 37.11 | 36.94 | 45.82 | 35.80 | 36.87 | 40.42 | 37.31 | 40.05 | 38.42 | 41.33 | 38.38 |
| + DCO | +18.83 | +13.48 | +14.66 | +15.91 | +12.74 | +16.33 | +15.32 | +19.62 | +19.85 | +16.26 | +10.56 | +17.47 | +19.42 | +11.64 | +17.89 | +11.23 |
| Llama3.2-3B | 43.00 | 47.98 | 43.92 | 38.72 | 37.89 | 32.10 | 34.73 | 44.14 | 34.21 | 32.74 | 38.04 | 33.17 | 39.14 | 37.21 | 38.65 | 35.67 |
| + DCO | +18.58 | +15.29 | +14.58 | +15.53 | +15.18 | +15.06 | +12.51 | +18.92 | +16.58 | +18.08 | +10.84 | +17.03 | +15.79 | +12.44 | +17.07 | +12.66 |

*Table 14.* Consistency improvements with English for each language across all models on BMLAMA.

| | EN | FR | NL | ES | RU | JA | ZH | KO | VI | EL | HU | HE | TR | CA | AR | UK | FA |
|---|---|---|---|---|---|---|---|---|---|---|---|---|---|---|---|---|---|
| **Accuracy** | | | | | | | | | | | | | | | | | |
| Qwen2.5-7B | 61.83 | 36.61 | 43.02 | 40.79 | 39.06 | 36.61 | 34.82 | 33.37 | 38.84 | 27.68 | 28.01 | 26.40 | 31.86 | 34.15 | 39.79 | 38.56 | 32.42 |
| + DCO | +5.61 | +19.31 | +14.57 | +15.68 | +12.61 | +16.57 | +14.23 | +17.75 | +15.35 | +12.61 | +12.56 | +13.89 | +12.73 | +13.23 | +9.32 | +13.34 | +10.88 |
| Qwen2.5-14B | 62.67 | 36.71 | 46.48 | 44.81 | 42.02 | 40.07 | 37.05 | 40.57 | 41.46 | 32.53 | 28.12 | 35.60 | 34.99 | 37.05 | 42.35 | 41.46 | 36.66 |
| + DCO | +6.33 | +21.10 | +12.90 | +12.56 | +13.67 | +15.18 | +11.89 | +14.28 | +15.85 | +13.12 | +14.62 | +13.51 | +13.34 | +12.89 | +11.17 | +15.12 | +13.51 |
| Gemma3-4B-pt | 66.07 | 32.42 | 43.86 | 39.68 | 35.21 | 30.30 | 26.34 | 32.48 | 36.83 | 31.81 | 30.69 | 29.91 | 34.60 | 32.42 | 31.58 | 38.90 | 30.08 |
| + DCO | +2.16 | +22.55 | +14.73 | +19.58 | +19.31 | +18.81 | +17.63 | +17.19 | +19.14 | +18.02 | +19.59 | +15.74 | +17.97 | +19.76 | +15.13 | +16.57 | +16.01 |
| Gemma3-12B-pt | 68.25 | 37.56 | 49.00 | 43.14 | 38.28 | 36.77 | 30.75 | 38.11 | 39.90 | 38.11 | 35.88 | 37.00 | 37.05 | 36.22 | 36.44 | 41.91 | 36.38 |
| + DCO | +1.55 | +21.09 | +13.05 | +17.57 | +17.47 | +17.30 | +16.91 | +16.58 | +19.03 | +18.11 | +18.47 | +14.23 | +19.53 | +17.52 | +13.84 | +18.02 | +12.62 |
| Qwen3-8B | 58.37 | 36.83 | 42.08 | 38.06 | 37.17 | 33.15 | 32.25 | 32.70 | 38.50 | 30.36 | 26.17 | 26.23 | 32.37 | 33.82 | 37.22 | 42.69 | 29.52 |
| + DCO | +6.90 | +16.69 | +13.11 | +14.28 | +12.10 | +14.67 | +12.95 | +13.17 | +12.34 | +11.55 | +15.79 | +11.94 | +13.05 | +14.51 | +9.43 | +9.65 | +11.33 |
| Qwen3-14B | 58.43 | 34.99 | 44.87 | 40.23 | 38.45 | 36.72 | 33.20 | 33.43 | 37.05 | 34.88 | 28.35 | 31.81 | 34.71 | 37.05 | 39.17 | 42.63 | 34.49 |
| + DCO | +8.07 | +18.86 | +12.66 | +13.84 | +13.84 | +16.01 | +16.30 | +14.78 | +16.69 | +17.35 | +16.91 | +11.77 | +14.17 | +14.85 | +12.00 | +11.39 | +13.05 |
| Aya-Expanse-8B | 67.02 | 45.31 | 46.82 | 52.73 | 35.21 | 36.22 | 37.28 | 34.04 | 49.55 | 32.87 | 22.32 | 33.37 | 35.83 | 32.76 | 37.61 | 39.79 | 33.15 |
| + DCO | +1.43 | +14.68 | +11.55 | +9.44 | +18.31 | +12.27 | +11.16 | +13.90 | +11.00 | +12.11 | +8.54 | +10.16 | +14.67 | +10.32 | +10.55 | +12.05 | +13.78 |
| Llama3.1-8B | 61.16 | 35.21 | 46.15 | 41.96 | 38.34 | 37.39 | 28.52 | 31.03 | 38.17 | 31.58 | 32.48 | 36.89 | 32.48 | 36.22 | 33.93 | 40.18 | 32.70 |
| + DCO | +7.17 | +23.61 | +14.73 | +17.58 | +17.80 | +14.34 | +22.43 | +18.19 | +21.26 | +19.87 | +17.58 | +9.82 | +18.41 | +20.64 | +12.50 | +18.36 | +14.73 |
| Llama3.2-3B | 61.05 | 35.38 | 43.08 | 38.67 | 34.60 | 32.31 | 24.67 | 26.95 | 35.55 | 29.35 | 28.74 | 35.16 | 27.57 | 35.21 | 32.25 | 36.22 | 28.24 |
| + DCO | +6.72 | +21.76 | +16.18 | +17.52 | +18.30 | +17.75 | +19.64 | +17.41 | +21.37 | +15.52 | +18.14 | +9.48 | +17.69 | +16.35 | +14.07 | +18.02 | +15.34 |

*Table 15.* Accuracy of each model on BMLAMA across all languages after DCO.

| Domain | AR | DE | ES | FR | HI | ID | IT | JA | KO | PT | SW | YO | ZH | BN |
|---|---|---|---|---|---|---|---|---|---|---|---|---|---|---|
| **Anatomy** | | | | | | | | | | | | | | |
| Base | 54.08 | 65.20 | 57.86 | 67.79 | 50.76 | 65.17 | 68.50 | 64.77 | 56.56 | 63.31 | 49.45 | 46.78 | 70.92 | 51.68 |
| + DCO | +13.10 | +7.02 | +17.56 | +12.04 | +9.30 | +13.69 | +11.42 | +11.12 | +10.49 | +19.53 | +4.81 | +3.20 | +12.00 | +7.91 |
| **Medical genetics** | | | | | | | | | | | | | | |
| Base | 66.91 | 78.76 | 75.42 | 71.46 | 56.92 | 74.52 | 74.54 | 77.25 | 66.76 | 77.88 | 62.25 | 59.95 | 79.97 | 62.66 |
| + DCO | +9.21 | +11.71 | +16.51 | +18.54 | +12.85 | +8.24 | +13.17 | +3.18 | +10.13 | +14.60 | +4.06 | +4.11 | +8.99 | +9.38 |
| **High school mathematics** | | | | | | | | | | | | | | |
| Base | 70.25 | 71.76 | 71.93 | 68.89 | 68.21 | 72.42 | 71.38 | 67.22 | 70.27 | 67.51 | 59.42 | 58.65 | 66.81 | 64.88 |
| + DCO | +8.59 | +8.55 | +14.72 | +13.07 | +10.86 | +13.30 | +11.21 | +12.76 | +9.48 | +16.02 | +8.12 | +4.15 | +14.26 | +12.67 |
| **College mathematics** | | | | | | | | | | | | | | |
| Base | 61.09 | 72.62 | 68.67 | 69.42 | 63.56 | 64.56 | 73.08 | 65.81 | 61.24 | 69.62 | 50.28 | 50.95 | 69.75 | 58.81 |
| + DCO | +13.72 | +7.71 | +17.22 | +13.31 | +8.55 | +14.01 | +10.30 | +8.80 | +10.38 | +13.61 | +12.08 | +7.26 | +13.39 | +8.68 |
| **High school world history** | | | | | | | | | | | | | | |
| Base | 83.50 | 83.06 | 87.21 | 86.67 | 74.73 | 81.53 | 87.19 | 83.20 | 54.06 | 85.53 | 56.25 | 44.98 | 41.79 | 74.35 |
| + DCO | +3.52 | +6.03 | +3.82 | +3.19 | +4.52 | +4.34 | +0.82 | +3.85 | +3.64 | +4.99 | -1.37 | +1.42 | +1.08 | +3.43 |

*Table 16.* Consistency with English under cross-domain settings using `Qwen2.5-14B`. The model is post-trained with data of 'high school microeconomics'.

| Domain | EN | AR | DE | ES | FR | HI | ID | IT | JA | KO | PT | SW | YO | ZH | BN |
|---|---|---|---|---|---|---|---|---|---|---|---|---|---|---|---|
| **Anatomy** | | | | | | | | | | | | | | | |
| Base | 68.89 | 43.70 | 46.67 | 45.93 | 53.33 | 34.07 | 50.37 | 51.85 | 57.04 | 44.44 | 43.70 | 32.59 | 31.85 | 72.59 | 40.00 |
| + DCO | +1.80 | 0.00 | +5.92 | +5.18 | +6.67 | +2.97 | +8.89 | +7.41 | 0.00 | +5.93 | +18.52 | -2.96 | -4.44 | -2.96 | +1.48 |
| **Medical genetics** | | | | | | | | | | | | | | | |
| Base | 82.00 | 61.00 | 73.00 | 66.00 | 66.00 | 46.00 | 74.00 | 63.00 | 79.00 | 59.00 | 69.00 | 53.00 | 53.00 | 78.00 | 53.00 |
| + DCO | +5.43 | +5.00 | +7.00 | +14.00 | +14.00 | +11.00 | +3.00 | +13.00 | -5.00 | +8.00 | +16.00 | +2.00 | -2.00 | +4.00 | +8.00 |
| **High school mathematics** | | | | | | | | | | | | | | | |
| Base | 53.70 | 43.70 | 45.56 | 39.63 | 45.19 | 43.33 | 42.96 | 39.26 | 48.52 | 47.04 | 43.70 | 34.81 | 30.74 | 60.00 | 40.74 |
| + DCO | +2.38 | +6.30 | +9.63 | +7.41 | +8.14 | +3.71 | +7.41 | +7.41 | +5.18 | +4.81 | +14.08 | +5.93 | +6.30 | -3.33 | +12.22 |
| **College mathematics** | | | | | | | | | | | | | | | |
| Base | 57.00 | 38.00 | 41.00 | 40.00 | 36.00 | 33.00 | 39.00 | 36.00 | 44.00 | 35.00 | 41.00 | 24.00 | 32.00 | 52.00 | 34.00 |
| + DCO | -0.36 | +11.00 | +15.00 | +8.00 | +22.00 | +10.00 | +11.00 | +18.00 | +16.00 | +11.00 | +13.00 | +15.00 | +4.00 | +10.00 | +9.00 |
| **High school world history** | | | | | | | | | | | | | | | |
| Base | 89.45 | 81.43 | 82.28 | 84.81 | 84.39 | 70.46 | 79.32 | 83.97 | 80.59 | 80.17 | 82.70 | 46.84 | 33.33 | 81.86 | 68.78 |
| + DCO | +0.55 | +1.27 | -0.85 | 0.00 | -0.85 | +0.43 | +1.69 | -1.69 | +1.27 | -2.95 | -0.42 | -2.96 | +0.85 | +0.42 | +2.53 |

*Table 17.* Accuracy under cross-domain settings using `Qwen2.5-14B`. The model is post-trained with data of 'high school microeconomics'.

