# OpenReview forum: "Post-Training Language Models for Crosslingual Consistency"
_ICML.cc/2026/Conference — ICML 2026 regular_

### Official Review · Reviewer_AQCL · 2026-03-05

**Soundness:** 3
**Presentation:** 2
**Significance:** 3
**Originality:** 2
**Overall Recommendation:** 3
**Confidence:** 4

**Summary:**

The author first formalized the cross-language consistency problem as a constrained reinforcement learning problem, provided a strict mathematical definition of consistency, derived the hyperparameter constraints and the optimal policy form for achieving consistency, and provided a complete theoretical framework for cross-language alignment.

**Compliance With Llm Reviewing Policy:**

Affirmed.

**Final Justification:**

The author's reply confirmed my assessment and I suggest keeping the score as it is.

**Key Questions For Authors:**

No Questions For Authors

**Limitations:**

yes

**Strengths And Weaknesses:**

Advantages: The proposed DCO algorithm does not require gold standard data, which compensates for the deficiencies of DPO (which relies on manual preference annotations), CALM (which is only applicable to multi-language scenarios and is susceptible to noise from low-resource languages), and also can be combined with DPO to form a "DPO + DCO" hybrid method to achieve the dual optimization of consistency and accuracy.

Disadvantages:
1）Although DCO is a consistency optimization method for models without explicit rewards, its core framework is entirely based on the preference optimization approach of DPO. It merely replaces the preference target from "human preference" to "cross-language distribution preference", without making any breakthrough innovations in the algorithm structure.
2) Although the paper conducted low-resource language analysis on Swahili and Yoruba, it only selected 2 low-resource languages, both of which were African languages. It did not cover low-resource languages in other regions such as Southeast Asia, Oceania, and the Americas, thus failing to prove the universality of DCO in low-resource languages.
3) In Section 6.5, the paper explores the application of DCO in open generation scenarios. However, it only uses Jaccard similarity to measure cross-language consistency, and does not employ more comprehensive open generation evaluation metrics (such as BLEU, CHRF, etc.). Additionally, the experiments only selected English-Chinese bilingual pairs and 2 models, and the training hyperparameters were not adequately optimized. The results are merely for feasibility verification and cannot prove the actual effectiveness of DCO in open generation scenarios. Moreover, the consistency performance in long text generation was not analyzed.
4) Although the paper mentioned that the code, training scripts and evaluation benchmarks would be released, it did not provide the details for reproducing the experiments. Some experimental results (such as Tab.1) have disordered table layout and unclear data annotations, which do not meet the requirements for experimental reproducibility of top-tier conference papers.

---

> ### Author Rebuttal · Authors · 2026-03-30
>
> We thank the reviewer for the feedback and comments. However, we found that certain key points of our paper may have been misunderstood, and we respectfully provide clarification below.
>
> ---
>
> **(1) On originality and the relation to DPO.**
>
> Firstly, we would like to clear up what appears to be a misconception about our paper’s contribution. The contribution we hope will last is our notion of (cross-lingual) consistency in alignment and our framing of the problem. Indeed, the DCO algorithm is nothing more than a means to an end. It is an efficient manner to (approximately) optimize our proposed objective. With respect to your point in (1), it is indeed inspired by DPO. This is because DPO was a great idea, and it helped us derive our own efficient approximation in a similar fashion.
>
> ---
>
> **(2) On universality of DCO and low-resource language coverage.**
>
> We would like to point out that the paper already evaluates DCO on **9 LLMs, 3 datasets, and 26 languages** overall, and the low-resource section was intended as a **targeted analysis** of direction control in especially difficult settings, rather than the sole evidence for multilingual generality. In §6.4, we specifically chose Swahili and Yoruba because they are among the hardest languages in our setup, and we show that DCO consistently improves crosslingual consistency under all weighting schemes, while EN-anchored settings yield the best accuracy/consistency tradeoff in these low-resource cases. The appendix further reports the same qualitative pattern for Yoruba.
>
> At the same time, to further strengthen this part, we tested the effectiveness of DCO on more low-resource languages, derived from an additional dataset, Global-MMLU. Specifically, **we add Amharic (am), Hausa (ha), Igbo (ig), Kyrgyz (ky), Malagasy (mg), Nepali (ne), Nyanja (ny), Sinhala (si), Shona (sn), Somali (so), and Telugu (te)** since they are low-resourece languages according to Joshi et al. (2020) taxonomy. In these additional experiments, DCO again yields significant consistency gains and substantial positive non-English accuracy gains. We will include these results in the revision.
>
> **Table: Improvements in consistency between low-resource languages with English**
> | Langs | am | ha | ig | ky | mg | ne | ny | si | sn | so | te |
> |---|---:|---:|---:|---:|---:|---:|---:|---:|---:|---:|---:|
> | Qwen2.5-14B|35.5|33.77|33.63|41.1|35.46|48.42|35.01|38.09|36.2|32.2|41.27|
> | +DCO |+8.30| +7.91 | +6.82 | +11.24 | +8.02 |+10.41|+9.19|+9.01|+7.91|+7.12|+10.27|
>
> **Table: Improvement in accuracy of low-resource languages**
> | Langs | en | am | ha | ig | ky | mg | ne | ny | si | sn | so | te |
> |---|---:|---:|---:|---:|---:|---:|---:|---:|---:|---:|---:|---:|
> | Qwen2.5-14B | 75.2 | 32.61 | 32.3 | 32.7 | 39.95 | 33.21 | 43.88 | 33.5 | 34.23 | 35.15 | 31.39 | 38.17 |
> | +DCO | -1.24 | +3.23 | +3.60 | +2.27 | +5.51 | +3.41 | +7.39 | +3.39 | +4.93 | +3.14 | +2.90 | +7.32 |
>
> ---
>
> **(3) On the open-generation results in §6.5.**
>
> We believe this criticism partly attributes a stronger claim to the paper than what is actually made. Section 6.5 is explicitly presented as **“Exploration: On-Policy RL Alignment”** and described as a **pilot experiment** with intentionally limited scope due to compute constraints. The goal of that section is not to claim a definitive benchmark for open-ended generation, but to test whether the same consistency-driven reward can be extended beyond the main probing-style setting into an on-policy RL setting. The paper already states that the EN–ZH setup, the two open-ended benchmarks, and the lightweight hyperparameter choices are meant as a feasibility check rather than an exhaustive study, and it explicitly leaves a more comprehensive investigation to future work.
>
> More importantly, the primary focus of our paper is **factual knowledge consistency**, where the consistency definition is especially well motivated because answers are objective and the candidate space is finite; this is also the setting assumed in the formalization and reflected in the main benchmarks. The open-generation section should therefore be read as a preliminary extension, not as the central empirical claim of the paper. In the revision, we will make this framing even clearer so that the intent of §6.5 cannot be misunderstood.
>
> ---
>
> **(4) On reproducibility and presentation.**
>
> We kindly argue that the paper contains reproducibility details. The paper already states that **all code, training scripts, and evaluation benchmarks are released**, and the main text explicitly points readers to **Appendix G** for training configurations. In addition, Table 1 already distinguishes methods trained with ground-truth answers using an asterisk. That said, we agree that the presentation can be improved further. In the revision, we will make the reproduction details easier to locate and improve the readability of Table 1, including clearer visual separation between methods that use gold labels and those that do not.

---

> > ### Author Rebuttal · Reviewer_AQCL · 2026-04-04
> >
> > We did not achieve the same results when replicating the code in the paper, and I remain skeptical about the experimental outcomes.

---

> > > ### Author Response · Authors · 2026-04-07
> > >
> > > We are confident that our results are fully reproducible.
> > >
> > > We attach an anonymized version of the complete code, data, and configuration files (including random seeds and dependency versions) needed to reproduce all experiments are available at https://anonymous.4open.science/r/anonymous_dco-0785/. The repository is well organized and includes step-by-step, one-click scripts for straightforward reproduction. We have also verified reproducibility across all results we reported on different versions of dependencies.
> > > All hyperparameters are clearly reported, including the hyperparameters for the online DCO experiments.
> > >
> > > To help us address the reviewer's concern, we would appreciate it if you could provide the benchmark, model, language, and settings that led to discrepant results, as well as any deviations of your implementation from the provided scripts. We are happy to assist in diagnosing the issue and would welcome the reviewer sharing their reproduction code or logs.

---

### Official Review · Reviewer_xBPY · 2026-03-11

**Soundness:** 3
**Presentation:** 2
**Significance:** 3
**Originality:** 2
**Overall Recommendation:** 4
**Confidence:** 4

**Summary:**

This paper targets cross-lingual knowledge inconsistency in multilingual LLMs, where similar questions asked in different languages can elicit contradictory answers. The authors propose Direct Consistency Optimization (DCO), a DPO-inspired RL method that uses a structured reward derived from the model itself, avoiding an explicit reward model. Experiments across multiple LLMs show DCO substantially improves cross-lingual consistency, outperforming prior methods when trained on multilingual samples and complementing DPO when gold labels are available.

**Compliance With Llm Reviewing Policy:**

Affirmed.

**Final Justification:**

This rebuttal has addressed my concerns.

**Key Questions For Authors:**

1. In Table 1, some configurations show non-trivial drops in English accuracy under DCO*. What conditions precipitate this trade-off?

2. Have you tested robustness when the anchor language’s belief is incorrect?

3. Could you report additional consistency metrics on MMMLU/XCSQA alongside RankC to triangulate findings for multiple-choice tasks?

**Limitations:**

yes

**Strengths And Weaknesses:**

## Strengths:

1. Introduces a principled, crosslingual reward that directly encodes consistency via other-language likelihoods and yields a product-of-experts optimal policy with a clear hyperparameter condition for consistency.

2. Offers a controllable alignment mechanism via direction weights, enabling practitioners to bias preservation of high-resource language performance while improving low-resource languages.

3. Evaluates on 9 diverse multilingual LLMs and 3 datasets covering 26 languages.

4. Includes an initial on-policy RL exploration for open-ended generation showing concurrent accuracy and consistency gains.

5. Clear problem setup, definition of crosslingual order-consistency, and derivation of the optimal policy; good use of figures/intuition and a straightforward loss definition for DCO.

6. Bridges preference-alignment paradigms and multilingual consistency with a simple, practically useful tool that can be adopted on top of existing models/training stacks.

## Weakness:

1. a more detailed algorithmic description (pseudo-code) of DCO would help reproducibility.

2. The role of parallel data is pivotal; clearer specification of how pairs (y_w, y_l) are sampled from candidates and handling of morphological variants or tokenization differences would be helpful.

3. The fairness of hyperparameter choices differs across methods; a sensitivity analysis or standardized budget would strengthen claims of superiority.

4. Low-resource languages are partly excluded in the joint setting and handled in a targeted section; more comprehensive coverage and analysis on scripts with known weaknesses (Indic, Perso-Arabic) would increase external validity.

---

> ### Author Rebuttal · Authors · 2026-03-30
>
> We thank the reviewer for the careful reading. We are glad that **the reviewer found the reward design principled, the optimal-policy derivation clear, and the empirical study meaningful**. We address the concerns below with additional experiments and analysis.
>
> ---
>
> **(1) Reproducibility / algorithmic description**
>
> We agree that §4.3 could benefit from a clearer algorithmic description. In the revision, we will add pseudocode that makes the DCO pipeline explicit. Due to the rebuttal length limit, we will provide the pseudocode in the next phase.
>
> ---
>
> **(2) Pair sampling / morphological variants / tokenization**
>
> In our current experiments, we deliberately use datasets with parallel candidate completions. As discussed in App.H, for MMMLU/XCSQA, the answer space is multiple-choice letters; for BMLAMA, the dataset provides parallel candidate words/entities across languages. Accordingly, we do not perform extra morphological normalization heuristics; we compute the model likelihood of the dataset-provided candidate completion in each language. For DCO training, MMMLU and BMLAMA firstly sample 5000 queries as the training set, with two randomly sampled parallel candidate completion pairs per query, while XCSQA contains only 1000 queries in total, thus, we randomly take 800 queries but repeats the sampling procedure of the candidate completions 6 times to obtain a comparable training size, resulting in a total of 5600 instances for the training. We will move the this to the main paper in the revision.
>
> ---
>
> **(3) Hyperparameter fairness.**
>
> We agree that the tuning protocol should be stated more explicitly. Our comparison is fair in the sense that each method is evaluated under its own best-tuned setting, rather than under a single shared learning rate, because SFT/DPO/CALM/DCO optimize different objectives and operate on different optimization scales. We report the final settings in Appendix G (SFT 5e-7, DPO 5e-6, CALM 5e-6, DCO 1e-5; Gemma on XCSQA 1e-6 to avoid overfitting), and we will highlight this protocol in the revision. In addition, §6.4 already provides a sensitivity analysis for DCO: across substantially different direction-weight (γ) settings, DCO consistently improves CLC, with predictable accuracy trade-offs rather than brittle behavior tied to one tuned configuration.
>
> ---
>
> **(4) Low-resource coverage / external validity.**
>
> We agree that broader low-resource coverage would strengthen the paper. To address this, we ran two additional analyses. First, we retrained Llama-3.1-8B with all MMMLU languages, including Swahili and Yoruba; average CLC improves from 60.1 to 72.0, while average accuracy is preserved (49.4 to 49.9). Second, **we test DCO on 11 extra low-resource languages (am, ha, ig, ky, mg, ne, ny, si, sn, so, te) in Global-MMLU**. On Qwen2.5-14B, DCO again yields strong consistency gains (+6.82 to +11.24) and substantial non-English accuracy gains (+2.27 to +7.39), with only a small English change of -1.24. See response (2) to Reviewer AQCL for the detailed numbers. We will include these results in the revision.
>
> ---
>
> **(Q1) Reason for the drop in English in Table 1**
>
> This trade-off mainly appears in the symmetric joint multilingual setting, where DCO is allowed to move all languages toward a shared multilingual consensus rather than explicitly preserving English. Our bilingual direction-control analysis supports this interpretation: with symmetric weights, CLC improves the most, but English may change more; when English is anchored, DCO still improves consistency while greatly reducing English answer churn. Thus, the trade-off is not intrinsic to DCO; it depends on the alignment direction and on how much movement is allowed in the stronger language.
>
> ---
>
> **(Q2) What if the anchor language is wrong?**
>
> DCO is consistency-oriented rather than correctness-oriented: unlike DPO, it does not require gold preferred/dispreferred labels, and the paired responses in Eq. (10) are randomly assigned rather than assumed to be intrinsically better or worse. As a result, if the anchor language is wrong, DCO can propagate that belief. Empirically, however, we find that DCO improves consistency with only marginal accuracy changes, and it works best as a complement to correctness-oriented post-training. Our practical recommendation is therefore to first improve correctness with labeled supervision (e.g., SFT/DPO), and then apply DCO to align responses across languages; when prior knowledge suggests one language is more reliable, the direction weights can be used to anchor that language.
>
> ---
>
> **(Q3) Additional consistency metrics**
>
> Besides RankC, we also evaluated Top-1 Overlap and COverlap (Jaccard over correctly answered query IDs). The trends are consistent with RankC across all tested models on both MMMLU and XCSQA. For example, on Qwen2.5-14B, DCO improves MMMLU by +12.6/+14.5/+14.8 and XCSQA by +6.8/+7.4/+6.3 on RankC/Top-1/COverlap respectively. Full tables will be provided in the next phase.

---

> > ### Author Rebuttal · Reviewer_xBPY · 2026-04-02
> >
> > I have no more question.

---

### Official Review · Reviewer_5hge · 2026-03-12

**Soundness:** 2
**Presentation:** 3
**Significance:** 3
**Originality:** 2
**Overall Recommendation:** 3
**Confidence:** 4

**Summary:**

The paper proposes Direct Consistency Optimization (DCO), a RL algorithm for LLMs to improve their cross-lingual knowledge consistency. By integrating an alignment reward with DPO, the optimization method is expected to achieve consistent answer distributions between language pairs.  Results show that DCO significantly improves the overall consistence and some non-English performances, while preserving the English performances. Analyses show a tradeoff between accuracy and consistency when adjusting the direction control parameters, and a pilot study shows that DCO can also be applied to online DPO.

**Compliance With Llm Reviewing Policy:**

Affirmed.

**Final Justification:**

I still think the comparison and analysis is not strong enough to support the claim in the paper.

**Key Questions For Authors:**

- In Figure 2 (right), **SW stable** yields higher consistency than **EN stable** and **Default**. Does it mean that changing English outputs are easier than changing non-English outputs? Why is this the case given the English-centered feature of the LLMs?

**Limitations:**

yes

**Strengths And Weaknesses:**

## Strengths
- The paper gives a practical definition of cross-lingual consistency in terms of consistent ranking in the output conditional distributions between language pairs, which sets a good reference for future work.

- The paper proposes the DCO reward and transforms it to the DPO style, which allows the integration of consistency and correctness awards through RL, yielding positive results.

- The paper takes the direction of alignment into account by setting direction controlling parameters, which is shown to be effective in controlling the stability of English performance while fostering cross-lingual consistency.

- The paper is well written, and the discussion part gives inspiring points about the possible mechanism and further use of the DCO method.

## Weaknesses

- The paper focuses on "cross-lingual knowledge consistency". However, since the dominant part of knowledge learninng is in pre-training, personally I am skeptical of whether DCO really changes the "knowledge base" inside the LLM. I think some probing tests will be beneficial to further support the generalizability claim.

- There are some other methods of cross-lingual consistency optimization, such as MAPO (She et al., 2024). Currently the compared baseline is SFT and DPO, both of which does not use the language consistency for assistant. I think the authors should compare DCO and these methods with results or discussions in the paper.

*She et al., MAPO: Advancing Multilingual Reasoning through Multilingual-Alignment-as-Preference Optimization, ACL 2024*

- Since the reward actually depend on the translation between two languages, it is still doutbful whether the quality of translation, especially when the input and output are long and complex, affects the consistency performance.

---

> ### Author Rebuttal · Authors · 2026-03-30
>
> We thank the reviewer for the careful reading and constructive feedback. We are pleased that **the reviewer recognizes the practical formulation of cross-lingual consistency, the effectiveness of DCO, and the value of our direction-control analysis**. Below, we clarify the main concerns.
>
> ---
>
> **(1) On whether DCO changes the model’s “knowledge base”**
>
> We would like to highlight that the paper already includes probing-style evidence through answer accuracy, particularly on the factual QA benchmark BMLAMA, which tests factual associations using prompts such as “The capital of the Netherlands is __”. As shown in Table 2, DCO yields substantial gains in both consistency and accuracy. For example, on Qwen2.5-14B, DCO improves BMLAMA by +15.4 in consistency and +14.2 in average non-English accuracy. These results suggest that DCO does not merely encourage more consistent outputs across languages but also improves the model’s ability to retrieve and express correct factual knowledge in non-English languages.
>
> ---
>
> **(2) On comparison with prior cross-lingual consistency methods**
>
> We would like to clarify that the paper already compares DCO against a label-free baseline that leverages internal cross-lingual agreement, namely CALM, in Table 1. As described in the paper, CALM applies DPO to encourage the model to prefer the majority-voted answer across languages, and we include it precisely because it is a direct baseline for cross-lingual consistency optimization in our setting. Empirically, Table 1 shows that DCO consistently outperforms CALM in overall cross-lingual consistency while better preserving accuracy. Meanwhile, we note that MAPO is not directly comparable to DCO, since it focuses on aligning sampled multilingual reasoning traces for reasoning tasks, whereas our work studies knowledge-intensive QA with finite candidate answers. For this reason, we view MAPO as a highly relevant related work and will discuss it in the revision.
>
> ---
>
> **(3) On the effect of translation quality**
>
> The main focus of our work is knowledge-intensive QA, where answers are mostly factual, objective, and drawn from a finite candidate set. In this setting, translation noise is considerably more limited than in free-form generation. To further examine this issue, we additionally evaluated DCO on more low-resource languages in Global-MMLU (Amharic, Hausa, Igbo, Kyrgyz, Malagasy, Nepali, Nyanja, Sinhala, Shona, Somali, and Telugu), where machine translation is more challenging. As shown below, even in this translation-unfriendly setting, DCO still yields **large consistency gains (approximately +6.8 to +11.2 points across languages), together with substantial non-English accuracy improvements (approximately +2.3 to +7.4 points)**, while English changes only modestly. We will add this discussion to clarify that DCO does not require perfect translation; rather, it remains effective as long as translation quality is reasonably sufficient for objective QA.
>
>
> **Consistency**
> |Langs|am|ha|ig|ky|mg|ne|ny|si|sn|so|te|
> |-|-:|-:|-:|-:|-:|-:|-:|-:|-:|-:|-:|
> |Qwen2.5-14B|35.5|33.77|33.63|41.1|35.46|48.42|35.01|38.09|36.2|32.2|41.27|
> |+DCO|+8.30|+7.91|+6.82|+11.24|+8.02|+10.41|+9.19|+9.01|+7.91|+7.12|+10.27|
>
> **Accuracy**
> |Langs|en|am|ha|ig|ky|mg|ne|ny|si|sn|so|te|
> |-|-:|-:|-:|-:|-:|-:|-:|-:|-:|-:|-:|-:|
> |Qwen2.5-14B|75.2|32.61|32.3|32.7|39.95|33.21|43.88|33.5|34.23|35.15|31.39|38.17|
> |+DCO|-1.24|+3.23|+3.60|+2.27|+5.51|+3.41|+7.39|+3.39|+4.93|+3.14|+2.90|+7.32|
>
> ---
>
> **(4) On Figure 2 (right), and whether English is easier to change**
>
> We do not interpret Figure 2 (right) as showing that English outputs are inherently easier to change than non-English outputs. In bilingual DCO, the relative amount of movement on each side is controlled by the direction parameters in Eqs. (8)–(9): A smaller γ keeps that language closer to the reference policy, while a larger γ imposes stronger cross-lingual alignment pressure on that side. Accordingly, the different regimes mainly determine **where the update is allocated**, rather than revealing an intrinsic asymmetry in “changeability.” Consistent with this interpretation, §6.4 shows that the low-resource side undergoes substantially more updates overall; for EN-SW, the EN-stable setting changes only 18.73% of English answers but 54.20% of Swahili answers, yielding the best accuracy–consistency trade-off. By contrast, the SW-stable setting shifts more of the update burden onto English (33.97% EN vs. 44.84% SW), achieving slightly higher consistency but at the cost of substantially worse English accuracy. Thus, the main takeaway is not that English is inherently easier or harder to edit, but that in our setting, English is the stronger channel and is therefore preferable to anchor during alignment.

---

> > ### Author Rebuttal · Reviewer_5hge · 2026-04-03
> >
> > The argument is not persuasive to me in comparison to other results. I don't thinking aligning reasoning trace (the method of MAPO as mentioned earlier) and aligning the output of answer (the task in this paper) are so different that they cannot be compared. A more important point is, the authors need to compare their method against previous attempt to align the output of different languages. The core point in CALM is that their take a majority voting from answers in different languages, while in MAPO their reply on the decision in English. Although the current paper present better results from CALM, I think the major source of improvement is the alignment to English, not the improvement in method itself. BTW, the results are not quite strong in comparison to DPO either.

---

> > > ### Author Response · Authors · 2026-04-07
> > >
> > > We thank the reviewer for this clarification. We now better understand the concern: whether the gains of DCO mainly come from anchoring to English, rather than from the method itself, and whether DCO is actually stronger than prior methods such as MAPO.
> > >
> > > **We have added a direct comparison with MAPO on MMMLU and GSM8K. Overall, DCO outperforms MAPO, and these results support that the gains are not explained solely by English anchoring.**
> > >
> > > ---
> > >
> > > First, we would like to clarify the supervision settings, which are important for interpreting Table 1.
> > > The asterisk (\*) denotes training with ground-truth answers. Thus, DPO\* uses gold answers, while DCO\* applies our consistency objective on top of a DPO-trained model as an ablation, which yields even higher consistency. In contrast, **DCO (without \*) does not use ground-truth answers and only requires paired parallel responses**. Therefore, comparing DCO directly with DPO* is not apples-to-apples. Despite this weaker supervision, DCO still substantially improves consistency and often accuracy.
> > >
> > > Second, regarding the concern that the main benefit comes from **alignment to English**, the multilingual joint training in Table 1 is not explicitly anchored to English: tuples $(q,c_1,c_2)$ are sampled from any two languages, and only **17% (4972/30000)** of DCO training instances include English. Thus, the gains cannot be attributed mainly to explicit English supervision.
> > >
> > > Third, we agree that MAPO is the most relevant prior baseline here, and we therefore added a direct comparison. Since MMMLU is a factual QA benchmark, we adapt MAPO’s translation-matching reward into a binary reward indicating whether the non-English answer matches the English answer. The results are shown below.
> > >
> > > Tab 1. MMMLU (Tab. 1 in the paper, Cont’d)
> > > |Method|Qwen2.5-14B|||Gemma3-12B-pt|||Qwen3-14B|||Aya-Expanse-8B|||Llama3.1-8B|||
> > > |-|-|-|-|-|-|-|-|-|-|-|-|-|-|-|-|
> > > ||$\rm CLC_{all}$|$\rm Acc_{EN}$|$\rm Acc_{Non}$|$\rm CLC_{all}$|$\rm Acc_{EN}$|$\rm Acc_{Non}$|$\rm CLC_{all}$|$\rm Acc_{EN}$|$\rm Acc_{Non}$|$\rm CLC_{all}$|$\rm Acc_{EN}$|$\rm Acc_{Non}$|$\rm CLC_{all}$|$\rm Acc_{EN}$|$\rm Acc_{Non}$|
> > > |DPO*|+12.3|+7.8|+13.9|+6.5|+1.8|+3.4|+3.0|+2.7|+4.2|+1.3|+2.5|+2.5|+10.1|+8.0|+8.8|
> > > |MAPO|+9.9 |+5.2|+4.9|+6.4|-1.7|-0.3|+3.9|-1.7|-0.6|+5.3|-1.1|+0.1|+4.1|+5.9|+3.1|
> > > |DCO|+10.6|+4.0|+9.6|+6.5|+0.9|+2.5|+2.7|+0.4|+1.3|+5.3|+0.5|+0.5|+9.4|+7.5|+7.6|
> > >
> > > Tab 2. Open generation on GSM8K (Tab. 4 in the paper, Cont’d)
> > > |Model|Acc EN|Acc ZH|Consistency|
> > > |-|-|-|-|
> > > |Qwen-2.5-7B-Instruct|89.2|83.6|84.7|
> > > |+MAPO|88.0|82.3|83.9|
> > > |+DCO|90.0|86.8|86.9|
> > > |Gemma-3-12B-it|90.3|87.1|87.7|
> > > |+MAPO|90.0|87.3|87.9|
> > > |+DCO|92.3|88.1|89.2|
> > >
> > > - **MMMLU**: DCO is overall stronger than MAPO on cross-lingual consistency, and usually yields larger accuracy gains in both English and non-English languages.
> > > - **GSM8K**: DCO outperforms MAPO in both consistency and accuracy.
> > >
> > > These results suggest that the gains of DCO are not merely due to using English as a reference. If that were the primary explanation, MAPO, which explicitly relies on the English decision, would be expected to match or exceed DCO. Empirically, this is not what we observe.
> > >
> > > More broadly, we believe the key distinction is the following. **MAPO transfers the English decision to other languages**, whereas **DCO directly optimizes a coupled multilingual objective** that encourages agreement across languages jointly. In this sense, DCO is not trying to copy English; it is learning a shared cross-lingual decision boundary. In the notation of the paper, DCO targets a multilingual consensus distribution
> > > $$\pi^*(y|x) = \prod_j \pi(\tau^j(y) | \tau^j(x) )$$, where j is the index of the language,
> > > whereas MAPO effectively anchors to the English distribution
> > > $$\pi^{MAPO}(y|x) = \pi(\tau^{en}(y) | \tau^{en}(x) )$$
> > >
> > > This difference matters in practice: as we show in Sec. 6.6, the multilingual consensus $\pi^*$ can achieve higher accuracy than any single-language distribution, including English. Intuitively, lower-resource languages tend to have higher-entropy predictions and therefore contribute less to the consensus, so they do not dominate or degrade the final decision.
> > >
> > > This also explains MAPO’s limited gains in our setting: MAPO helps most when English substantially outperforms non-English, but our base models already have much smaller English/non-English gaps. This is consistent with GSM8K, where MAPO brings little or no gain, while DCO still improves consistency and accuracy. Overall, the new results show that DCO is stronger than MAPO under the same base models, and its gains cannot be explained simply by "relying more on English". Instead, the coupled multilingual objective itself matters.

---

### Official Review · Reviewer_SH9k · 2026-03-13

**Soundness:** 4
**Presentation:** 3
**Significance:** 3
**Originality:** 4
**Overall Recommendation:** 5
**Confidence:** 4

**Summary:**

The paper addresses the issue of crosslingual consistency (CLC) in multilingual LLMs. The authors propose Direct Consistency Optimization (DCO), a method inspired by reinforcement learning (RL) that aligns the likelihood of answers across parallel prompts without requiring an explicit reward model. For example, if the likelihood of candidate A being preferred over candidate B is true in language 1, this preference should also hold true in language 2.
In experiments on 9 LLMs across 26 languages on three benchmarks (MMMLU, XCSQA, BMLAMA), the authors show that DCO significantly improves CLC. They also demonstrate that training on one domain transfers well to improvements in out-of-domain subjects.

**Compliance With Llm Reviewing Policy:**

Affirmed.

**Key Questions For Authors:**

How exactly were the "translation-based pseudo pairs" formed and scored in the open-ended solution traces for GSM8K in Section 6.5? Besides the higher computational effort, do you expect this method to scale well to longer answers?

The underlying assumption of Crosslingual Consistency (CLC) is that the model should provide the exact same factual preference regardless of language. Have you considered how DCO behaves on queries involving culturally subjective or localized knowledge, where a "consistent" answer might actually represent an English-centric bias overriding a valid regional perspective?

**Limitations:**

There is no explicit Limitations section, but the Impact Statement can probably be counted as this. It states:
“This paper presents work whose goal is to advance the field of machine learning. There are many potential societal consequences of our work, none of which we feel must be specifically highlighted here”

Why not? What are the consequences the authors have in mind? One problem that should be highlighted is that enforcing strict crosslingual consistency risks overwriting localized, language-specific cultural contexts with a monolithic (likely English-centric) worldview. Additionally, the assumption that exact translational mappings between languages exist should also be discussed as a fundamental limitation of the approach.

**Strengths And Weaknesses:**

Strengths:
The paper presents a solid mathematical foundation for its methods, and the empirical evaluation is well-designed. The authors test on multiple distinct models and sizes (ranging from 4B to 14B parameters, including Llama, Qwen, Gemma, and Aya) across varied knowledge retrieval tasks, ensuring their claims are not model- or task-specific. Showing that the method generalizes well on out-of-domain tasks demonstrates that this approach does not merely lead to overfitting on the training data.

Weaknesses:
The theoretical formulation relies heavily on the assumption that exact translational mappings exist between candidate answers across languages. This assumption is not always true. Especially when scaling this method to open-ended answers or language-specific questions, one would expect problems (e.g., “Do rain and train rhyme?” is likely false in most languages besides English). That said, this assumption should hold for the factual benchmarks tested here.

---

> ### Author Rebuttal · Authors · 2026-03-30
>
> We thank the reviewer for the **positive comments of our technical contribution, experimental comprehension, and out-of-domain generalization**. We would like to address the two concerns below, regarding (i) scaling to open-ended generation and (ii) culturally-/regional diverse knowledge.
>
> ---
>
> **(1) Open-ended GSM8K setup / pseudo pairs.**
>
> In Section 6.5, we construct the translation-based pseudo pairs as follows. We first let the model generate two free-form solution traces in English for the same GSM8K question. We then translate the question and both traces into the target language using a lightweight translator (Qwen3-4B), yielding parallel tuples of the form ($q^{en}, r_1^{en}, r_2^{en}$) and ($q^ℓ,r_1^ℓ,r_2^ℓ$). These parallel tuples are then used for online DCO training, exactly as described in Section 6.5. Importantly, DCO is correctness-agnostic: we do not use a reward model or gold labels to score the traces. Instead, the training signal comes from aligning relative preferences across the translated pairs. This makes the method applicable whenever approximately parallel responses are available, whether from pre-translated datasets or from on-the-fly translation with a small auxiliary model.
>
> That said, we view this online DCO experiment as a preliminary exploration rather than a claim that DCO is fully solved for unrestricted open-ended generation. In principle, the approach extends to longer answers, but in practice, longer solution traces increase token cost and are more susceptible to translation noise and semantic drift. Our main focus in this paper, therefore, remains factual knowledge settings, where answers are objective and typically drawn from a finite candidate set.
>
> ---
>
> **(2) Existence of exact translational mappings**
>
> Thank you for raising this interesting question. We believe it is useful to distinguish between two cases of difficult translational mapping.
>
> (i) First, some questions may be difficult to translate literally, while the intended answer should remain the same across languages. This is exactly the case for the reviewer’s example, “Do rain and train rhyme?” If a user asks this question in another language, they may still be curious about the relationship between the exact English words “rain” and “train,” rather than about their translated counterparts. In that case, a faithful translation should preserve the English lexical items, e.g., French: *Est-ce que « rain » et « train » riment ?*; Arabic: *هل تتناغم كلمتا "rain" و "train"؟*; Chinese: *“rain” 和 “train” 押韵吗？* Under such a translation, the expected answer remains the same across languages. Therefore, in this category, the primary issue is the translation capability of difficult sentences, which could be mitigated by using a stronger translator.
>
> (ii) Second, some questions can have different expected answers across regions, cultures, or local contexts, even when the translation itself is fully faithful. For instance, the question “Which side of the road do cars drive on?” would typically be answered, “right” in most English-speaking contexts, whereas its Japanese counterpart may instead call for “左側” (“left”). We would discuss more in the following section (3).
>
> ---
>
> **(3) Limitation Paragraph: Culturally subjective or localized knowledge.**
>
> We agree that this is an important limitation. Our formulation explicitly assumes translational mappings in factual QA settings where answers are objective and the candidate space is finite; this is the regime targeted by our benchmarks and theory. For culturally subjective, localized, or perspective-dependent queries, enforcing cross-lingual consistency may indeed be undesirable, because the ideal answers could be diverse. In such cases, people from different regions will get undesirable answers if we blindly improve knowledge consistency between English and their language. Therefore, we agree that this is a situation that needs careful consideration, and we shouldn’t expect to use DCO to enforce a single universal worldview, but rather as a tool appropriate for objective factual settings. That said, these cases are beyond the scope of the current paper and therefore do not diminish our main contributions. We will explicitly discuss it in the Limitations section in the revision.

---

> > ### Author Rebuttal · Reviewer_SH9k · 2026-04-03
> >
> > My points have been addressed.

---

### Decision · Program_Chairs · 2026-04-30

**Decision:**

Accept (regular)

**Comment:**

The paper addresses the issue of crosslingual consistency (CLC) in multilingual LLMs. The authors propose Direct Consistency Optimization (DCO), a method inspired by reinforcement learning (RL) that aligns the likelihood of answers across parallel prompts without requiring an explicit reward model. For example, if the likelihood of candidate A being preferred over candidate B is true in language 1, this preference should also hold true in language 2. In experiments on 9 LLMs across 26 languages on three benchmarks (MMMLU, XCSQA, BMLAMA), the authors show that DCO significantly improves CLC. They also demonstrate that training on one domain transfers well to improvements in out-of-domain subjects.

Strengths
The authors test on multiple distinct models and sizes (ranging from 4B to 14B parameters, including Llama, Qwen, Gemma, and Aya) across varied knowledge retrieval tasks.
The proposed DCO algorithm does not require gold standard data
The paper gives a practical definition of cross-lingual consistency in terms of consistent ranking in the output conditional distributions between language pair

Weaknesses
The authors need to compare their method against more previous attempts to align the output of different languages. More comparison has been prepared for the rebuttal and should be added to the paper.
The underlying assumption of Crosslingual Consistency (CLC) is that the model should provide the exact same factual preference regardless of language. This is not always true and should be discussed in the paper and experiments on culture relevant benchmarks would improve the paper.